# Code-driven Number Sequence Calculation: Enhancing the inductive Reasoning Abilities of Large Language Models

## Abstract

Large language models (LLMs) make remarkable progress in reasoning tasks. Among different reasoning modes, inductive reasoning, due to its better alignment with human learning, attracts increasing interest. However, research on inductive reasoning faces certain challenges. First, existing inductive data mostly focuses on superficial regularities while lacking more complex internal patterns. Second, current works merely prompt LLMs or finetune on simple prompt–response pairs, but do not provide precise thinking processes nor implement difficulty control. Unlike previous work, we address these challenges by introducing *CodeSeq*, a synthetic post-training dataset built from number sequences. We package number sequences into algorithmic problems to discover their general terms, defining a general term generation (GTG) task correspondingly. Our pipeline generates supervised finetuning data by reflecting on failed test cases and incorporating iterative corrections, thereby teaching LLMs to learn autonomous case generation and self-checking. Additionally, it leverages reinforcement learning with a novel Case-Synergy Solvability Scaling Reward based on both solvability, estimated from the problem pass rate, and the success rate of self-directed case generation, enabling models to learn more effectively from both successes and failures. Experimental results show that the models trained with *CodeSeq* improve on various reasoning tasks and can preserve the models' OOD performance. Our code and data are available at: https://anonymous.4open.science/r/CodeSeq2-1ABE.

## 1 Introduction

Recent groundbreaking advances in natural language processing (NLP), such as `OpenAI-o3` (Pfister & Jud, 2025), `Claude-Sonnet-4` (Benzon, 2025) and `DeepSeek-R1` (DeepSeek-AI, 2025), make remarkable progress in reasoning capabilities of large language models (LLMs) (Franceschelli & Musolesi, 2023; Jin et al., 2024; Xi et al., 2023; Xu et al., 2024).

Existing reasoning paradigms can be categorized into deductive reasoning (Li et al., 2025c) and inductive reasoning (Lu, 2024). The former, including mathematical (Ahn et al., 2024; Chen et al., 2024b) and code reasoning (Jiang et al., 2024a; Liu et al., 2023), utilizes general principles to achieve specific conclusions logically. It has already been extensively studied in recent years (Lu et al., 2024; Wang et al., 2024b). In contrast, the latter (Han et al., 2024) involves drawing general conclusions from specific observations. Given that this inductive mode is key to knowledge generalization and better aligns with human learning, it is relatively essential and thus attracts increasing interest.

However, research on inductive reasoning in LLMs still faces certain challenges. **I. Missing complex patterns.** Current inductive reasoning data, such as List Functions (Li et al., 2025b) or ARC (Wang et al., 2024c), contains only superficial formal regularities among observations (Qiu et al., 2024), failing to form complex internal patterns. **II. Not properly trained.** Many studies merely prompt closed-source LLMs (He et al., 2025; Li et al., 2025a) or finetune on simple prompt–response pairs (Lee et al., 2025), but they do not provide precise thinking processes nor implement difficulty control. Hence, they can not fundamentally enhance the inductive capabilities of trainable LLMs.

To address **challenge I.**, we novelly employ number sequences as the source of inductive reasoning data, which requires finding the general terms based on the given number series. Number sequence

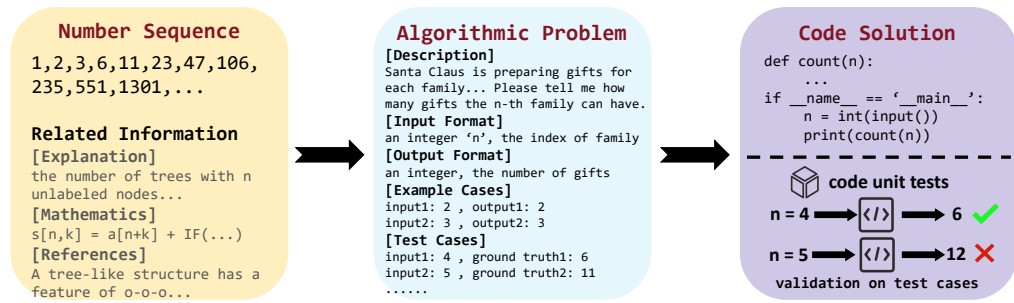

Figure 1: We package the number sequences into algorithmic problems with the help of their related information. Although a story description is used for packaging, we ensure each algorithmic problem takes the index of a sequence term as input and outputs the number corresponding to that index (if the input is 2, what we actually want is to output the 2-nd term of the sequence). Therefore, the code solution is the computational process of the number sequence's general term. LLMs generate the code solution by means of the example cases (usually the first few terms, which help with thinking), and we can then verify the correctness of the solution on the test cases through code unit tests.

problems not only focus on superficial changes in numbers, but also reveal underlying patterns across terms, thereby better reflecting LLMs' inductive abilities (more explanations in Appendix A.3). As for **challenge II.**, we believe the core of inductive reasoning lies in *validating inductive hypotheses with cases*. Therefore, we build a number sequence synthetic data (Bauer et al., 2024) pipeline, forming a post-training dataset (SFT and RL) denoted as *CodeSeq*. This post-training dataset aims to train LLMs to engage in a human-like case-based reasoning process, thus enhancing their inductive capabilities. Correspondingly, a general term generation (GTG) task is proposed to assess such capacity. Considering LLMs' weak representation ability for pure numbers (Marjieh et al., 2025; Zhou et al., 2024) and the inherent difficulty in expressing sequence general terms through mathematical formulas (Barry, 2012; Zhang, 2022), we package number sequences into algorithmic problems to find the general term of each sequence through a code solution (Figure 1). In this way, we skillfully represent many number sequence general terms, which are difficult to express directly in mathematical notations, with code. Then code unit tests (Yang et al., 2024a; Hui et al., 2024) can be used to verify such solutions across number cases in the original sequence (Havrilla et al., 2024).

Our synthetic data pipeline consists of three main parts. **(1) Sequence Algorithmization.** We scrape and filter thousands of number sequences with their related information from websites. We then leverage a working agent to package each number sequence into an algorithmic problem, accompanied by two example cases. Another guiding agent directly generates the solution based on the problem description and the inputs of the example cases to validate whether the algorithmic problem description meets the example cases, thus further ensuring the correctness of the problem itself. We divide the validated problems into two groups, preparing them separately for supervised finetuning (SFT) and reinforcement learning (RL) data construction. **(2) Case-based Reflection Injection.** The working agent generates code solutions for the first group. We verify whether the code solution holds for all test cases via code unit tests. The guiding agent provides modification suggestions for the failed test cases and asks the working agent to make corrections iteratively. We inject this reflection process into the Chain-of-Thought (CoT) and form the SFT data, so that LLMs can learn to generate cases autonomously and perform self-checking. **(3) Solvability-Estimated Selection.** For the second group, each problem is sampled multiple times to measure its pass rate, providing a solvability estimation. To ensure the model's learnability, we pick up those problems that could only be solved correctly after multiple rollouts. As a result, we construct the RL training data. Meanwhile, we design a Case-synergy Solvability Scaling Reward based on the solvability and the success rate of autonomous case generation, to guarantee a controllable learning process and further improve the model's proficiency in self-directed case generation.

To verify the effectiveness of *CodeSeq*, we train two LLMs, applying the SFT data followed by the RL data. Experimental results show that the models tuned with *CodeSeq* not only perform well on the in-domain GTG task, but can also be generalized to close-domain code tasks. Moreover, our models maintain their comprehensive reasoning abilities in out-of-domain (OOD) scenarios.

In summary, the main contributions of this paper are listed as follows:

- To our knowledge, this is the first work to utilize number sequences as the training data and study their impact on LLMs regarding the inductive abilities. We package number sequences into algorithmic problems to find the general terms. Hence, many general terms, which are difficult to express directly in mathematical notations, can be represented by code solutions.

- We propose a number sequence synthetic data pipeline, forming a post-training dataset *CodeSeq* that consists of SFT and RL data. The SFT data is organized by reflecting the code solution on failed cases and iteratively making corrections, teaching LLMs to learn autonomous case generation and self-checking. What's more, we use the pass rate as an estimation of solvability to construct the RL data and raise a Case-synergy Solvability Scaling Reward to facilitate controllable learning and further improve the model's proficiency in self-directed case generation.

- Our synthetic data *CodeSeq* is proven effective for various reasoning tasks and can preserve the models' OOD performance, demonstrating the potential of such data on inductive reasoning.

## 2 RELATED WORK

### 2.1 INDUCTIVE REASONING

LLM Reasoning can be categorized into deductive reasoning (Johnson-Laird, 1999; Li et al., 2025c) and inductive reasoning (Hayes et al., 2010; Lu, 2024). Deductive reasoning, such as well-defined tasks like mathematical reasoning and code reasoning (Lu et al., 2024; Wang et al., 2024b), utilizes general principles and axioms to achieve specific goals, pursuing logical certainty. While inductive reasoning is quite the opposite. It involves drawing general conclusions from specific observations, which is the most universal and essential approach in knowledge discovery (Han et al., 2024). Thus, the inductive reasoning of LLMs attracts increasing interest among researchers.

Most existing works (Li et al., 2025b; Wang et al., 2024c; Li et al., 2025d) on inductive reasoning in LLMs use datasets such as List Functions or ARC (Hammond et al., 2024), which only focus on superficial regularities among observations without constructing deeper underlying patterns. To improve the inductive abilities of models, current approaches merely prompt LLMs (He et al., 2025; Li et al., 2025a; Liu et al., 2024) to iteratively generate hypotheses or train with simple samples (Lee et al., 2025). To address these issues, we novelly employ number sequences as the source of inductive reasoning data and build a synthetic data pipeline to provide high-quality training data.

### 2.2 CODE GENERATION

Code serves as a crucial link between humans and machines. Code programs are marked by several notable traits: precision, logical structure, modular design, and maintainability (Sun et al., 2025; Wan et al., 2023). In the era of AI, code generation mainly goes through three stages: (1) code embedding (Girdhar et al., 2023), (2) code pretrained models (Wang et al., 2023), and (3) code generation in LLMs. These three stages have corresponding relationships with the development of NLP. The most prominent feature of code generation is learning with execution feedback (Yang et al., 2023). It enables compilers or interpreters to produce accurate feedback on cases automatically. This process can also be called the code unit tests (Le et al., 2022; Ma et al., 2025), which typically runs in a manually written, isolated sandbox environment (Park et al., 2024).

In the era of LLMs, there are three main methods for enhancing code generation ability: (1) decoding-enhanced, that is, using methods such as self-planning (Jiang et al., 2024b), self-filling (Martínez-Magallanes et al., 2023), and Program of Thought (PoT) (Bi et al., 2024) to improve generation accuracy. (2) feedback-driven, which is similar to tree search (Dainese et al., 2024; Matute et al., 2024) and applies unit tests to provide supervised signals. (3) natural-language (NL) guidance (Wang et al., 2024a), which means deploying natural language to guide the generation process of code.

In this paper, considering LLMs' weak representation ability for pure numbers and the inherent difficulty in expressing general terms via intuitive mathematical notations, we package number sequences into algorithmic problems to find general terms through code solution, proposing a GTG task for LLMs to evaluate the inductive reasoning ability correspondingly. We construct SFT and RL data with Case-based Reflection Injection and Solvability-Estimated Selection separately.

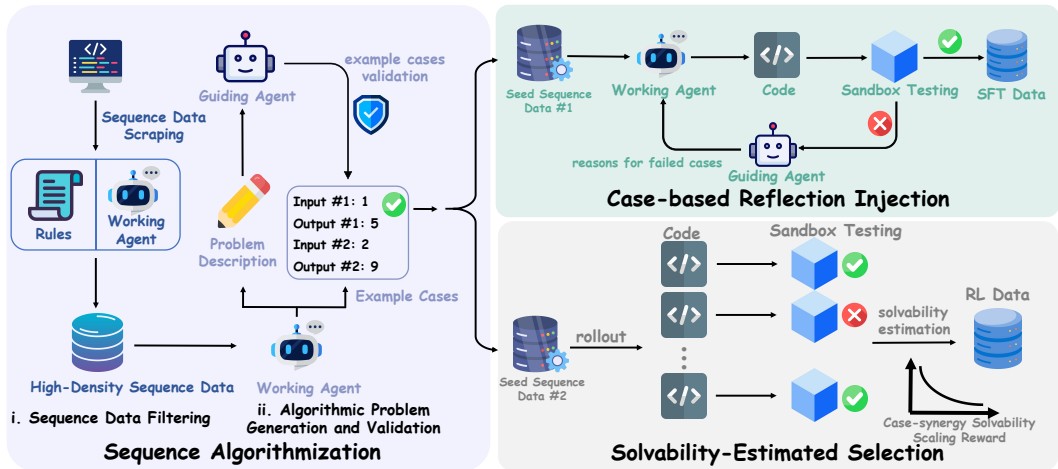

Figure 2: The pipeline of number sequence synthetic data. Sequence Algorithmization performs algorithmic problem generation and verifies the correctness of the problems. Case-based Reflection Injection generates SFT training data through case-based reflection and incorporates the process into CoT. Solvability-Estimated Selection estimates problem solvability using pass rates and selects RL training data accordingly. These two parts together form the post-training dataset for our *CodeSeq*.

## 3  NUMBER SEQUENCE SYNTHETIC DATA PIPELINE

In this paper, we employ number sequences as the source of inductive reasoning data and design a three-step synthetic data pipeline in Figure 2: Sequence Algorithmization (Section 3.1), Case-based Reflection Injection (Section 3.2), and Solvability-Estimated Selection (Section 3.3). For more detailed information on the principle, purpose, and prompt usage of each step, see Appendix A.4.

### 3.1  SEQUENCE ALGORITHMIZATION

Sequence algorithmization refers to obtaining, filtering, and transforming the number sequence data into algorithmic problems, as well as verifying the correctness of the problems.

**Sequence Data Filtering**   We scrape many number sequences and their related information from various websites. The related information includes the source, formula, general term description, and so on. Since we plan to package the sequences into algorithmic problems by a powerful LLM working agent (Guo et al., 2025; Wang et al., 2025), our first need is to judge whether sequences have sufficient relevant information to be successfully packaged. We first manually write rules to remove sequences with insufficient information, such as those without the calculation process or evolve from other sequences (requiring additional webpage links for references). Then we prompt the working agent to self-plan (Jiang et al., 2024b) the steps for generating an algorithmic problem and self-reflect (Wang et al., 2024d) on whether each step contains enough information. The above operations result in a batch of sequences with high information density. See more in Appendix A.4.1.

**Algorithmic Problem Generation and Validation**   We next have the working agent generate an algorithmic problem about the general terms for each high-density sequence, along with two example cases. Example cases provide the standard input and output cases for this algorithmic problem to help the problem solvers understand it better. To further verify the correctness of the algorithmic problems themselves, we utilize another powerful LLM as a guiding agent (Guo et al., 2025). We feed the problem description and the two example cases' inputs into the guiding agent, which produces the results directly. By comparing these results with the outputs in the generated example cases, we can determine whether the current problem description matches the example cases, thus verifying the correctness of the problem. In this way, we obtain a series of well-formatted and correct number sequence algorithmic problems, which we refer to as seed sequence data. We divide it into two groups, preparing them separately for SFT and RL data construction. More details are in Appendix A.4.2.

### 3.2 Case-based Reflection Injection

In this part, we introduce the construction of SFT data that incorporates a case-driven reflection reasoning process, enabling the LLMs to determine automatically whether the current code solution is correct and learn the general thinking paradigm of inductive reasoning (Appendix A.4.3).

After obtaining the seed sequence data, we let the working agent directly generate the code solutions for the algorithmic problems in the first group. Since the problem description involves a natural language story about the general term of a number sequence, the code solution represents the computational process of the general term. Like example cases, we manually select 5 to 7 items from the original number sequence randomly as test cases to validate the correctness of the code solution.

Imitating previous code unit tests (Hui et al., 2024), we use test cases to test the correctness of each code solution in an isolated sandbox environment. If a code solution fails on a test case, we ask the guiding agent to reflect and provide the reason for the failure. We then give that reason, along with the failed test case, back to the working agent, who regenerates the code solution iteratively.

Ultimately, through continuous reflections (Huang et al., 2024; Wang et al., 2024d) and modifications, we achieve a code solution that passes all the test cases. We inject this case-driven reflection process into the CoT. To ensure data diversity, we perform multiple samplings of both the problem descriptions and the initial attempt code solutions, thereby constructing the SFT training data, named $CodeSeq_{\text{SFT}}$.

### 3.3 Solvability-Estimated Selection

This part introduces our method for estimating the solvability of algorithmic problems based on pass rates to filter RL training samples. We see the pass rate as an estimation of solvability. For assigning different rewards to a model depending on problem solvability, and further improving the model's proficiency in self-directed case generation, a new Case-synergy Solvability Scaling Reward (CSSR) is proposed to assist the RL training process. In this way, we hope to improve the upper limit of inductive reasoning further. More introductions are in Appendix A.4.4.

We perform $N$ rollouts on each algorithmic problem in the second group, using a model with a similar number of parameters as the base LLMs to be trained. This allows us to estimate the difficulty of the current problem as accurately as possible. The greater the difficulty of a problem, the lower its solvability for the current model. For each sampled code solution of the same problem, we also use sandbox verification to determine whether it passes all test cases. Suppose there are $N_p$ code solution samples that pass all test cases, then the estimation of the solvability $\hat{Sov}$ can be expressed as:

$$\hat{Sov} = \frac{N_p}{N} \tag{1}$$

To maximize the model's potential while ensuring learnability, we mostly pick up those sequences that could only be solved correctly after multiple rollouts. In other words, we choose those algorithmic problems that could only be solved correctly after more than one try, which implies that they have an inferior solvability for the LLMs, as the RL training data.

In the RL training process, we design a new reward function CSSR:

$$\mathcal{R} = \begin{cases} -1, & \text{if formatted error} \\ 0, & \text{if any test case fail to execute} \\ -\lambda \log(\hat{Sov} + \epsilon) + (1-\lambda)\frac{N_{tc}}{N_c}, & \text{if all test cases pass} \end{cases} \tag{2}$$

where $\epsilon$ is used to prevent the reward from becoming infinitely large when solvability approaches 0. $N_c$ and $N_{tc}$ denote the number of cases generated by the model's self-reflection in CoT and the number of correctly generated cases among them, respectively.

In the CSSR formula, if a code solution has an incorrect format, the reward is set to $-1$. If any of the test case fails, the reward is set to 0. If one code solution passes all test cases, the reward is divided into two components. The first part is the scaling of solvability, in which the logarithmic function is used for smoothing, stabilizing the model's learning and convergence: the lower the solvability, the higher the reward is. The second part represents the success rate of autonomous case generation in CoT. $\lambda$ is employed to balance the two components. We refer to this part of the data as $CodeSeq_{\text{RL}}$.

| $CodeSeq_{\text{SFT}}$ | # sample | # pattern | Max. R tokens | # R sample | Avg. rounds | Max. rounds |
|---|---|---|---|---|---|---|
| | $5,487$ | $1,271$ | 4.5K | $3,348$ | 2.56 | 5 |

| $CodeSeq_{\text{RL}}$ | # sample | # pattern | Max. R tokens | Min. Sov | Max. Sov | Avg. Sov |
|---|---|---|---|---|---|---|
| | $1,543$ | $1,543$ | 1.5K | 0.00 | 0.46 | 0.28 |

Table 1: Data statistics of our post-training datasets *CodeSeq*. '# sample', '# pattern', and 'Max. R tokens' mean the number of training samples, the number of sequence patterns, and the maximum response length, respectively. '# R sample' represents the number of training samples with case-based reflection, for which we record the average and maximum reflection rounds.

### 3.4 SYNTHETIC DATA STATISTICS

Table 1 shows the data statistics of *CodeSeq*, which demonstrates the diversity of our data. The SFT and RL data together encompass about $3,000$ number sequence patterns, capturing the underlying rules that govern how each number sequence general term is formed, and thus reflecting diverse real-world inductive patterns. The model can enhance its inductive reasoning ability through such a rich variety of patterns. The maximum response length for the SFT and RL training samples is 4.5K tokens, which can almost be adapted to the training of all open-source models in the academic community. In the SFT data, it has an average of about 2.56 reflection rounds. This proves that we effectively incorporate case-based reflection signals into the number sequence inductive reasoning data. In the RL data, we select the majority of algorithmic problems with solvability in the range of $(0, 0.46]$. This approach ensures the difficulty of getting the right code solutions, while also maintaining the potential for learnable possibilities. Other details and statistics are in Appendix A.5.

## 4 EXPERIMENTS

To prove the effectiveness of our synthetic data *CodeSeq*, we employ it to perform SFT and RL on existing open-sourced LLMs. We test its performance on the in-domain GTG task, three close-domain code reasoning benchmarks, and three out-of-domain comprehensive reasoning benchmarks.

### 4.1 TRAINING IMPLEMENTATION

We conduct training on two widely used LLM backbones: `LLaMA3-8B-Instruct` (Grattafiori et al., 2024) and `Qwen2.5-7B-Instruct` (Qwen et al., 2025). To maintain the models' instruction-following (Zhu et al., 2024) and other inherent abilities when SFT, we mix $CodeSeq_{\text{SFT}}$ with the latest post-training corpus Tülu 3 (Lambert et al., 2025). During the RL stage, to preserve the general capabilities of the LLMs as much as possible, we train the two models on $CodeSeq_{\text{RL}}$, applying the GRPO algorithm (Ramesh et al., 2024). To avoid randomness, we train the models with different seeds and take the average. About backbones and training parameters are in Appendix A.6.

### 4.2 BENCHMARKS AND EVALUATION

To evaluate the improvement in inductive reasoning ability of the two LLMs after training, we construct 200 new algorithmic problems to serve as our test set, which is also validated via the GTG task. Since the number sequences are converted into code problems, we consider code tasks to be close-domain and utilize three related benchmarks: Humaneval (Chen et al., 2021), MBPP (Austin et al., 2021), and LiveCodeBench (Jain et al., 2024). Meanwhile, we also deploy three out-of-domain comprehensive reasoning benchmarks to measure the general capabilities: MMLU (Hendrycks et al., 2021), BBH (Suzgun et al., 2022), and GaoKaoBench (Zhang et al., 2024b). Finally, we utilize OpenCompass (Contributors, 2023) to evaluate the results. More information is in Appendix A.6.

### 4.3 MAIN RESULTS

Table 2 exhibits the results of `LLaMA3-8B` and `Qwen2.5-7B` on seven benchmarks after training with *CodeSeq*. We can draw the following conclusions. (1) Both base models significantly improve the in-domain GTG task after training with *CodeSeq*. This explains the backbones' neglect of GTG

| | in-domain | close-domain | | | out-of-domain | | |
|---|---|---|---|---|---|---|---|
| **Models** | GTG | Heval | MBPP | LCBench | MMLU | BBH | GaoKao |
| `InternLM2.5-7B` | 14.74 | 67.68 | 61.86 | 16.11 | - | - | - |
| `InternLM2.5-20B` | 15.82 | 73.78 | 68.43 | 17.95 | - | - | - |
| `DeepSeek-R1-7B` | 27.87 | 84.39 | - | 16.66 | - | - | - |
| `DeepSeek-R1-32B` | 29.75 | 85.22 | - | 24.33 | - | - | - |
| `LLaMA3-8B` | 9.27 | 59.15 | 63.81 | 11.26 | 62.23 | 46.40 | 45.80 |
| w/ *CodeSeq*$_\text{SFT}$ | 13.44$_\uparrow$ | 65.24$_\uparrow$ | 65.79$_\uparrow$ | 15.14$_\uparrow$ | 62.34$_\uparrow$ | 47.44$_\uparrow$ | 45.48$_\downarrow$ |
| w/ *CodeSeq*$_\text{RL}$ | 40.35$_\uparrow$ | 62.80$_\uparrow$ | 67.29$_\uparrow$ | 14.33$_\uparrow$ | 62.42$_\uparrow$ | 46.14$_\downarrow$ | 45.62$_\downarrow$ |
| w/ *CodeSeq* | 44.22$_\uparrow$ | 65.85$_\uparrow$ | 68.45$_\uparrow$ | 16.14$_\uparrow$ | 63.19$_\uparrow$ | 48.15$_\uparrow$ | 45.70$_\downarrow$ |
| Δ | +34.95 | +6.70 | +4.64 | +4.88 | +0.96 | +1.75 | −0.10 |
| `Qwen2.5-7B` | 26.89 | 82.31 | 71.59 | 41.97 | 75.49 | 64.80 | 75.75 |
| w/ *CodeSeq*$_\text{SFT}$ | 36.42$_\uparrow$ | 83.45$_\uparrow$ | 73.93$_\uparrow$ | 43.42$_\uparrow$ | 74.96$_\downarrow$ | 64.48$_\downarrow$ | 76.72$_\uparrow$ |
| w/ *CodeSeq*$_\text{RL}$ | 63.36$_\uparrow$ | 83.73$_\uparrow$ | 73.51$_\uparrow$ | 42.33$_\uparrow$ | 75.12$_\downarrow$ | 64.89$_\uparrow$ | 76.81$_\uparrow$ |
| w/ *CodeSeq* | 69.55$_\uparrow$ | 86.13$_\uparrow$ | 75.49$_\uparrow$ | 43.89$_\uparrow$ | 75.56$_\uparrow$ | 66.60$_\uparrow$ | 77.46$_\uparrow$ |
| Δ | +42.66 | +3.82 | +3.90 | +1.92 | +0.07 | +1.80 | +1.71 |
| w/o Tülu 3. (SFT) | 34.33 | 81.55 | 73.14 | 37.67 | - | - | - |
| w/ just Tülu 3. (SFT) | 25.92 | 80.03 | 71.48 | 41.87 | - | - | - |
| w/o Reflection. (SFT) | 27.33 | 80.78 | 72.08 | 42.13 | - | - | - |
| w/o Sov. (RL) | 30.60 | 82.31 | 72.82 | 41.92 | - | - | - |
| w/o CaseSucc. (RL) | 62.95 | 83.07 | 73.03 | 41.83 | - | - | - |
| w/o CSSR. (RL) | 61.89 | 81.55 | 72.94 | 41.75 | - | - | - |

Table 2: Results of `LLaMA3-8B-instruct` and `Qwen2.5-7B-instruct` on our GTG test set and other six benchmarks after training with *CodeSeq*. 'Δ' indicates the performance difference between the trained models and the base models. The lower part of the table presents the results of ablation studies based on `Qwen2.5-7B-instruct`, where '(SFT)' and '(RL)' stand for the model only undergoing the SFT and RL process, respectively.

and other similar inductive reasoning tasks, as well as the great potential of using synthetic data for inductive reasoning. (2) *CodeSeq* enables the two models to achieve good transfer performances on close-domain code reasoning tasks. This is mainly attributed to our number sequence synthetic data being framed as algorithmic problems, incorporating inductive-based thinking, such as reflection based on cases during the problem-solving process. This phenomenon, to some extent, demonstrates that inductive reasoning is the key to knowledge generalization. (3) *CodeSeq* does not compromise the models' OOD performances. Although there is a performance drop on three comprehensive reasoning benchmarks, the decrease is no more than 0.55 points (`Qwen2.5-7B` with *CodeSeq*$_\text{SFT}$ on MMLU), reflecting the robustness of our data. Our goal is merely to prove that *CodeSeq* does not degrade, rather than improve the OOD performances. Therefore, we do not evaluate the baselines on the OOD benchmarks. (4) Base models trained with *CodeSeq* could achieve performance in GTG and certain code tasks comparable to models with three times more parameters. In the GTG task, our small-parameter models significantly outperform `InternLM2.5-20B` and `DeepSeek-R1-32B` after training. Meanwhile, for code tasks, *CodeSeq* enables `LLaMA3-8B` to surpass `InternLM2.5-20B` on MBPP, and allows `Qwen2.5-7B` to outperform `DeepSeek-R1-32B` on HumanEval.

## 4.4 ABALTION STUDIES

To demonstrate the effectiveness of our synthetic data, we conduct ablation studies separately during the SFT and RL stages in Table 2 (below). (1) 'w/o Tülu 3.' and 'w/ just Tülu.' express training without Tülu 3 and just training on Tülu 3 during SFT, separately. Case studies show that Tülu 3 helps the model retain its instruction-following ability, but using such data alone does not improve the model's performance on GTG and code reasoning tasks. (2) 'w/o Reflection.' means that during SFT, CoT without case-based reflection is used. This leads to a significant performance drop on the GTG task, indirectly demonstrating that this type of CoT is beneficial for inductive reasoning. (3) 'w/o

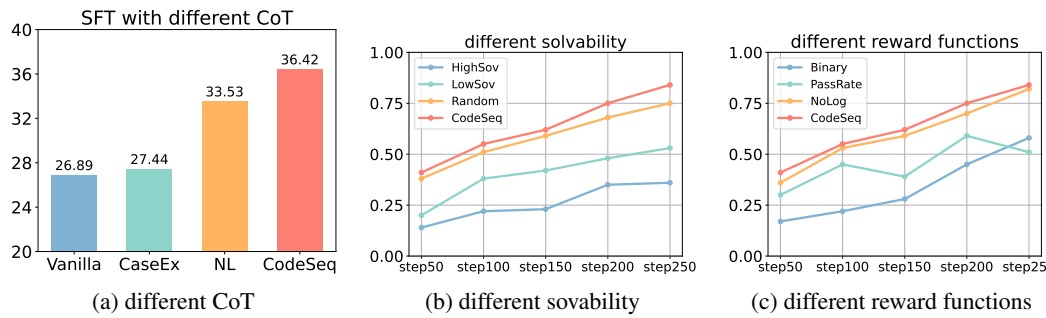

(a) different CoT        (b) different sovability        (c) different reward functions

Figure 3: Results on different training settings with `Qwen2.5-7B-Instruct`.

Sov.' and 'w/o CaseSucc.' speak for directly removing the first and the second term in Formula 2 separately, which both lead to a drop in model performance, denoting the importance of these two components. We can clearly observe that the solvability term plays a more important role in the reward function. (4) 'w/o CSSR.' signifies just using the 'Binary' reward function to train the RL data. This also cannot achieve the best performance. We will elaborate on this in Section 4.6.

## 4.5 COMPARISON WITH TOP CLOSED-SOURCE LLMS

We compared `Qwen2.5-7B-Instruct` with three top closed-source LLMs on the GTG and the ARC task, which is also an inductive reasoning task, in Table 3. Results show that after training with CodeSeq, the 7B model can significantly narrow the performance gap with top large-parameter models on the GTG task, and also reveals a certain degree of generalization on the OOD ARC task.

## 4.6 EFFECTS OF DIFFERENT TRAINING SETTINGS

We adopt different training settings (descriptions of each setting are in Appendix A.7.4) to validate the effectiveness of our synthetic data in Figure 3 on `Qwen2.5-7B-Instruct`. (1) We investigate the impacts of different CoT formats on SFT. 'Vanilla' means the result without training. Our case-based reflection CoT ('CodeSeq') outperforms both the methods of just providing partial number sequence terms ('CaseEx') and presenting the general terms purely through natural language guidance ('NL') during reflections. (2) We explore the impacts of training samples with different solvability during RL. We define 0 to 0.3 as low solvability ('LowSov') and 0.7 to 1.0 as high solvability ('HighSov'), partitioning the original training data accordingly while keeping the

|  | **GTG** | **ARC** |
|---|---|---|
| `DeeepSeek-R1` | 62.23 | 15.8 |
| `o3-mini-high` | 73.20 | 34.5 |
| `Claude-Sonnet-4` | 75.67 | 23.8 |
| `Qwen2.5-7B` | 26.89 | 0.0 |
| w/ *CodeSeq* | 69.55 | 0.9 |

Table 3: Comparison with top closed-source LLMs on GTG and ARC task.

number of training samples fixed. Results demonstrate that the solvability ranged from 0 to 0.46 ('CodeSeq'), which indicates training with moderately low-solvability samples yields the highest reward during the RL process. (3) 'Binary', 'PassRate', and 'NoLog' represent the following, respectively: assigning 1 only if all test cases pass, otherwise 0; just using the solvability in Formula 1; and not using the logarithmic function in Formula 2. Results in the figure and ablation studies both indicate that our specially designed CSSR reward function is the optimal choice.

## 4.7 THE SCALING BEHAVIORS OF *CodeSeq*

We investigate several scaling laws (Isik et al., 2024; Kaplan et al., 2020) introduced by *CodeSeq* for the two models. The results are shown in Figure 4. (1) We first examine how training samples with different reflection rounds affect model performance in the GTG task using `Qwen2.5-7B-Instruct`. '0r' represents the base model. We ensure the same amount of SFT training data for different reflection rounds. As the number of reflection rounds increases, the model's performance on the GTG task improves, but with diminishing returns. This suggests that utilizing more challenging samples

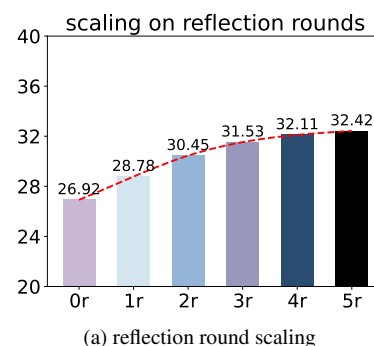 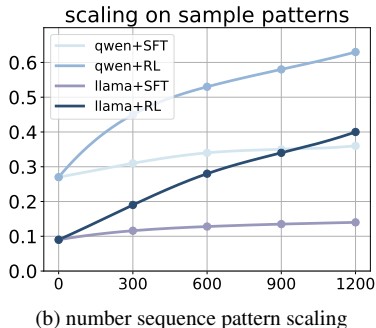

(a) reflection round scaling

(b) number sequence pattern scaling

Figure 4: Scaling results on the number of reflection rounds and the number of sequence patterns.

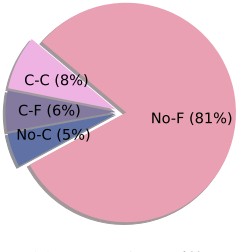 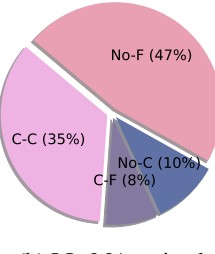 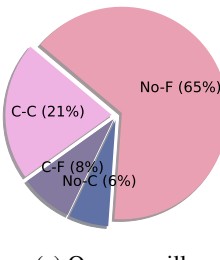 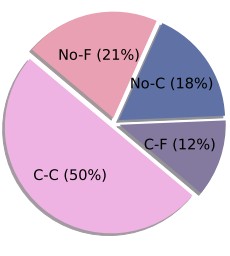

(a) LLaMA-vanilla

(b) LLaMA-trained

(c) Qwen-vanilla

(d) Qwen-trained

Figure 5: We study the relationship between case generation in CoT and prediction accuracy. 'C-C' and C-F': when all CoT cases exist in the original number sequence, the code solution is correct or false; 'No-C' and 'No-F': when at least one case is missing, the code solution is correct or false.

(with more reflection rounds) is more beneficial for SFT. (2) We also dig the scaling behavior of the diversity of number sequence patterns. The accuracy on the GTG task improves as the number of sequence patterns increases with the two backbones. Results indicate that more patterns help the models raise their inductive reasoning ceiling. More explanations are in Appendix A.7.7.

## 4.8 CAN MODELS LEARN TO REASON WITH CASES?

We investigate the relationship between LLMs' success rate of autonomous case generation and the accuracy of the code solution, thereby determining whether models can reason with cases. We quantify this relationship in Figure 5. Results imply that *CodeSeq* improves both the models' success rate in constructing cases and the overall problem-solving accuracy ('C-C'). These two are positively correlated, indicating that this case-based way of reasoning is feasible. More explanations about how to think with cases are shown in Appendix A.7.8.

## 5 CONCLUSIONS

In this paper, we study the impact of inductive reasoning data on LLMs. We novelly employ number sequences as the source of inductive reasoning data and build a synthetic data pipeline, forming a post-training dataset denoted as *CodeSeq*. We package number sequences into algorithmic problems to find the general terms through code solutions, proposing a general term generation (GTG) task. We construct supervised finetuning data by reflecting on the failed cases and iteratively making corrections, teaching LLMs to learn autonomous case generation and self-checking. We use the pass rate as an estimate of solvability to construct the reinforcement learning data, and propose a Case-synergy Solvability Scaling Reward based on both solvability and the success rate of self-directed case generation to facilitate learning. Experimental results show that the models trained with *CodeSeq* improve on various reasoning tasks and can preserve the models' OOD performance.

## ETHICS STATEMENT

Our data is constructed using LLMs, focusing on the scientific task of number sequences. We ensure correctness through a rigorous verification process, and there are no security concerns involved throughout the entire pipeline. We obtain all the synthetic data with API Keys through a paid subscription. The entire process and outcomes are free from intellectual property and ethical legal disputes, incorporating ethical considerations.

## REPRODUCIBILITY STATEMENT

We provide links to the code and data during the review process. We will also package the code and data into a zip file as the supplementary materials in OpenReview. Once our paper is accepted, we will release all relevant materials publicly on GitHub.

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

# A APPENDIX

## A.1 THE USE OF LARGE LANGUAGE MODELS (LLMS)

In this paper, we primarily use LLMs for data synthesis, model training, and evaluation. During the writing process, we rely on LLMs to correct errors and warnings in Overleaf. In other words, we mainly apply LLMs to assist us in writing with the LaTeX language.

## A.2 PRELIMINARY EXPERIMENTS

Number sequences are an excellent type of data for inductive reasoning because deriving the general term formula of a number sequence requires inferring an abstract and universal representation based on the specific terms of the number sequence.

After the process in Section 3, we obtain the SFT and RL number sequence synthetic data. In addition to the post-training dataset and the test set, we further construct 200 samples as a prior test set and conduct the next number prediction experiments. The next number prediction can be viewed as a simplified version of the GTG task, making it easier to assess how well existing LLMs understand number sequences and to quantify their inductive reasoning abilities. We conduct these experiments with the three most powerful LLMs in terms of reasoning ability: `o3-mini-high`, `Claude-Sonnet-4`, and `DeepSeek-R1`. We provide each model with the first few terms of a number sequence and require it to directly output the next term.

We use the following prompt in Figure 6 to have LLMs predict the next number in the given sequence, and the results are in Table 4.

> **The Prompt for Next Number Prediction**
>
> I will give you a sequence now.
> Please predict the next number based on the terms I provide.
> Please respond in JSON dictionary format: {"thought": xxx, "answer": xxx},
> where the "thought" section represents your inductive reasoning process for the sequence, and the "answer" section should directly give a number representing the final predicted answer.
>
> \<bos\> (the sequence) \<eos\>

Figure 6: The prompt for the next number prediction task.

| | |
|---|---|
| o3-mini-high | 38.0 |
| Claude-Sonnet-4 | 43.0 |
| DeepSeek-R1 | 32.0 |

Table 4: Results of the next number prediction task for the three top close-sourced LLMs.

The results show that even the most powerful existing LLMs still fail to achieve high accuracy on this task, indirectly reflecting their weakness in inductive reasoning ability.

## A.3 NUMBER SEQUENCES

Although some recent works (Chen et al., 2024a; Cheng et al., 2024; Wang et al., 2024c) begin to explore inductive reasoning, there still remain some challenges. One of the biggest challenges is that existing inductive data mostly focus on superficial regularities while lacking more complex internal patterns. Commonly used data, such as List Functions or ARC, only emphasizes the superficial correspondences between observations. These correspondences consist of transformations

of shapes, element substitutions, local pattern similarities, and so on. They focus more on enabling models to discover immediately visible 'pattern matches' rather than deep logical rules. Number sequence data, although superficially just a series of consecutive numbers, often contain complex mathematical formulas or generative rules that cannot be directly inferred from their appearance. Such data is inherently clean and noise-free, ensuring correctness. Furthermore, sequence data is highly interpretable, with strong inductive relationships between elements, making it well-suited for constructing supervised synthetic data. Lastly, superficial format similarities (Chen et al., 2024a; Cheng et al., 2024) or toy/game-like inductive signals (Wang et al., 2024c) in other inductive tasks are not essential enough for real-world applications. In contrast, our number sequence-based approach enables natural transfer to tasks like math or code reasoning, offering strong practical utility.

**Why number sequences?**

Compared to other inductive reasoning data, number sequences exhibit excellent scalability and high verifiability, and thus can be leveraged to produce synthetic data automatically. Other types of data are often difficult to construct and require human labor.

(1) Most relational datasets rely on a limited and fixed set of operations (e.g., map/reduce/sort). Because the underlying rules are highly enumerable, models tend to memorize template-like mappings rather than truly learning inductive reasoning. (2) Linguistic induction is highly sensitive to context, and multiple answers may be equally valid. Pure semantic matching is neither as rigorous nor as convenient as executable code for verification. (3) Visual inductive problems are feasible, but as the complexity increases (e.g., in terms of the number of constraints or the scale of variables), the cost of expansion and verification rises significantly.

Compared with other inductive tasks, number-sequence tasks exhibit deeper structural complexity and a more abstract pattern-representation capability in inductive reasoning.

First, the regularities in number sequences often involve multi-level recursions or nonlinear operations, resulting in a depth of pattern structure that far exceeds pixel-based transformations or simple logical mappings in ARC tasks. Second, the generative rules of number sequences can form complex inductive structures through combinations of operations, periodic nesting, and hierarchical dependencies, requiring models not only to capture explicit patterns but also to internalize the underlying generative mechanisms.

LLMs perform well on short-term pattern recognition but show a significant drop in accuracy when dealing with complex recursions and closed-form inference, indicating that number-sequence tasks are more challenging at the abstract reasoning level.

In short, the essence of inductive reasoning is discovering patterns. Compared with other data that exhibit only a single form of regularity, number sequences feature nested and progressive patterns, making them more complex and closer to real-world scenarios.

Current inductive datasets, as typical LLM reasoning sources, are likely to suffer from data leakage—that is, the model may have already seen similar content during training, preventing further improvement in capability. Therefore, they cannot enhance OOD inductive reasoning ability. For example, the Intern series, as well as the Minimax-M1 and Seed-thinking models, all make extensive use of large amounts of induction-style data, such as puzzles and strings. Moreover, numerous studies have shown that training on previously seen samples can cause overfitting and harm a model's OOD capabilities.

The above discussion further illustrates that, in order to improve a model's reasoning ability while maintaining OOD performance, it is crucial to construct high-quality data from new data sources. This highlights the effectiveness of our data and methods.

**Why do number sequences have practical use?** (1) Although number sequences typically appear in everyday life within pedagogical settings, in our work, they merely serve as a starting point. Our main goal is to construct training data for inductive reasoning. Most existing LLM synthetic reasoning datasets and benchmarks focus on deductive reasoning (e.g., math and code), while the training data for inductive reasoning is relatively scarce. The main reason is that current data for inductive reasoning heavily relies on manual expert intervention, and the reasoning process is difficult to express in natural language. Our proposed sequence-based pipeline can automatically generate synthetic data for inductive reasoning, thereby filling this gap. (2) Inductive reasoning is more aligned

with how humans learn and can also enhance deductive reasoning capabilities. Therefore, it is a fundamental skill for LLMs. (3) The number sequence task aligns with real-world applications because it comes from the sequences, patterns, and dependencies found in the real world. Solutions of the sequence problems can be directly applied to fields such as finance, technology, and healthcare, meeting human demands for automation, efficiency, and accuracy. Overall, due to the scalability, interpretability, verifiability, and practicality of number sequences, they serve as an excellent source of data. (4) Recent LLM training efforts increasingly focus on non-deductive reasoning tasks, such as toy/puzzles/chess tasks (Giadikiaroglou et al., 2024), symbolic reasoning (Sullivan & Elsayed, 2024), and abstract logical reasoning (Malkinski & Mandziuk, 2025). While these tasks may not be as prevalent as mathematical reasoning, they are equally important for evaluating the overall reasoning ability of LLMs, and they are all fundamentally based on inductive thinking. (5) Our dataset has already been used in the training of two famous open-source LLMs in 2025, one at the 8B scale and one exceeding 200B (we will disclose these names in the camera-ready version). In the training of these two models, our data is evaluated on over 10 reasoning benchmarks, resulting in an average improvement of 2.1 points and nearly 1 point, respectively. Therefore, it holds much practical value.

**Can the number sequence task truly reflect a model's inductive reasoning ability?** Yes, they can. From a mathematical perspective, computing the general term of a number sequence requires the model to summarize patterns and regularities from the sequence's terms, which aligns with the process of inductive reasoning. At the same time, we also test our data via one standard inductive reasoning benchmark (ARC), and the experimental results in Table 3 demonstrate that our data possesses generalization.

**More explanations related to the real-world scenarios.** (1) Existing works lack real-world data and are primarily based on manually crafted or automatically constructed synthetic data. Current inductive reasoning studies also lack evaluation in real-world scenarios. (2) In real-world natural language scenarios, (i) there is a large amount of redundant information, sentence perplexity is relatively high, and ambiguity may arise with strong dependence on context; (ii) the 'patterns' are weakly expressed, making it difficult for LLMs to learn and induce specific rules or conclusions effectively. (3) Therefore, this paper uses number sequence synthetic data for dataset construction, model training, and performance evaluation to fill the above gaps. (i) Most previous works focus on toy or abstract synthetic reasoning scenarios, which fail to provide precise thinking or reflection processes for LLMs to learn from. (ii) Compared to other synthetic data, number sequences meet human demands for automation, efficiency, and accuracy. Overall, due to the scalability, interpretability, verifiability, and practicality of number sequences, they serve as an excellent source of data.

## A.4 SUPPLEMENTARY INFORMATION FOR SYNTHETIC DATA PIPELINE

In this section, we provide more detailed information and additional examples to clarify the number sequence synthetic data pipeline. For the working agent, considering the need to make frequent calls, and for cost-saving purposes, we chose `DeepSeek-V3`[1] (DeepSeek-AI et al., 2024), while for the guiding agent, we select one of the most powerful reasoning models, `o3-mini-high` (Pfister & Jud, 2025)[2], so that the reflection process will be more accurate. We will demonstrate how these strong instructions-following agents work under the guidance of prompts with detailed instructions. In fact, small models can also accomplish this task; they just require more attempts to obtain qualified data. Larger models enable more efficient completion of these tasks.

The data in this paper consists of inputting a number sequence and outputting the general term of the sequence. However, since the general term of the sequence cannot be fully described using pure mathematical language, and we aim to construct interpretable and verifiable thinking processes for LLMs to learn, we package the number sequences into algorithmic problems. Ultimately, we input an algorithmic problem and output the corresponding code solution and the thinking process.

**How to prevent potential bias?**

Our pipeline minimizes bias as much as possible.

(1) Length bias. The length of a sequence is independent of the data: each term in the number sequence corresponds to one case in the algorithmic problem (Figure 1). We ensure that every

---

[1]https://chat.deepseek.com/

[2]https://openai.com/index/openai-o3-mini/

problem contains exactly two example cases and 5 to 7 test cases, and this quantity is strictly controlled.

(2) Pattern bias. Each number sequence corresponds to a distinct inductive pattern. We ensure that each pattern appears an equal number in the training or test sets, preventing the base model or the test distribution from being skewed toward any particular pattern.

### A.4.1 Sequence Algorithmization: Sequence Data Filtering

We scrape a large number of number sequences and their related information mainly from three math-related websites: OEIS[3], Euler[4], and Fenbi[5].

**OEIS** OEIS is currently the most comprehensive website for number sequences. Due to the diversity and complexity of its data, over 80% of our data comes from it.

**Euler** Project Euler is a challenge-based platform that integrates mathematics and programming, well-suited for learners who prefer practical engagement. It offers a wide range of problems related to number sequences. This portion accounts for approximately 10% of the data.

**Fenbi** Fenbi is a Chinese platform for national civil service examinations, featuring a large number of multiple-choice questions on number sequence patterns. These questions require the calculation of decimals and fractions, while we only need number sequence problems with integers. Therefore, the proportion of data from Fenbi is relatively small.

Each page on the website corresponds to a number sequence and all its information, including the source, formula, general term description, and so on. Normally, each number sequence has a unique identifier, such as 'ABCD'. The URL for the number sequence on the website takes the form 'https://oeis/ABCD', so we can conveniently scrape these sequences via their URLs. We give an example of one OEIS webpage in Figure 7.

We need to filter the information for each candidate number sequence to ensure the accuracy of the algorithmic problem generation process. We first manually write rules to filter out number sequences with insufficient information, including: (1) those with too few terms, which will result in any powerful agent being unable to understand the mathematical logic of the sequence thoroughly. (2) those that evolve from other sequences, which will result in people being unable to scrape enough information about the current sequence from the existing website (requiring additional webpage links for references). (3) those without 'mathematics' or 'programming' fields. This is for the working agent to initially filter information, making it easier to generate the code problems.

We then prompt the working agent to self-plan the steps for generating a problem and self-reflect (Wang et al., 2024d) on whether each step contains enough information. This prompt is shown in Figure 8. The above operations result in a batch of number sequences with high information density.

### A.4.2 Sequence Algorithmization: Algorithmic Problem Generation and Validation

We next have the working agent generate an algorithmic problem about the general term for each high-density number sequence, along with two example cases. The prompt for problem generation is in Figure 9. Example cases provide the standard input and output formations of this algorithmic problem to help the problem solvers understand it better. We also present an example of a number sequence algorithmic problem in Figure 10.

To further verify the correctness of the algorithmic problems, we utilize another powerful LLM as a guiding agent. We feed the problem description and the two example cases' inputs into the guiding agent, which produces the results directly (prompt in Figure 11). By comparing these results with the outputs in the generated example cases, we can determine whether the current problem description matches the example cases, thus verifying the correctness of the problem. Take the algorithmic

---

[3]https://oeis.org/

[4]https://projecteuler.net/

[5]https://www.fenbi.com/spa/tiku/guide/home/xingce/xingce

The OEIS is supported by the many generous donors to the OEIS Foundation.

**THE ON–LINE ENCYCLOPEDIA OF INTEGER SEQUENCES ®**

founded in 1964 by N. J. A. Sloane

[Search] [Hints]

(Greetings from The On-Line Encyclopedia of Integer Sequences!)

A054924    Triangle read by rows: $T(n, k)$ = number of nonisomorphic unlabeled connected graphs with n nodes and k edges ($n \geq 1$, $0 \leq k \leq n(n-1)/2$).

1, 0, 1, 0, 0, 1, 1, 0, 0, 0, 2, 2, 1, 1, 0, 0, 0, 0, 3, 5, 5, 4, 2, 1, 1, 0, 0, 0, 0, 0, 6, 13, 19, 22, 20, 14, 9, 5, 2, 1, 1, 0, 0, 0, 0, 0, 0, 11, 33, 67, 107, 132, 138, 126, 95, 64, 40, 21, 10, 5, 2, 1, 1, 0, 0, 0, 0, 0, 0, 23, 89, 236, 486, 814, 1169, 1454, 1579, 1515, 1290, 970, 658, 400, 220, 114

(list; graph; refs; listen; history; text; internal format)

OFFSET    1,11

REFERENCES    R. W. Robinson, Numerical implementation of graph counting algorithms, AGRC Grant, Math. Dept., Univ. Newcastle, Australia, 1976.

LINKS    R. W. Robinson, Rows 1 to 20 of triangle, flattened (corrected by Sean A. Irvine, Apr 29 2022)
G. A. Baker et al., High-temperature expansions for the spin-1/2 Heisenberg model, Phys. Rev., 164 (1967), 800-817.
Sean A. Irvine, Java code (github)
Gordon Royle, Small graphs
M. L. Stein and P. R. Stein, Enumeration of Linear Graphs and Connected Linear Graphs up to p = 18 Points. Report LA-3775, Los Alamos Scientific Laboratory of the University of California, Los Alamos, NM, Oct 1967

EXAMPLE    Triangle begins:
1;
0,1;
0,0,1,1;
0,0,0,2,2,1,1;
0,0,0,0,3,5,5,4,2,1,1;
0,0,0,0,0,6,13,19,22,20,14,9,5,2,1,1;
the last batch giving the numbers of connected graphs with 6 nodes and from 0 to 15 edges.

MATHEMATICA    A076263 gives a Mathematica program which produces the nonzero entries in each row.
Needs["Combinatorica`"]; Table[Print[row = Join[Array[0&, n-1], Table[ Count[ Combinatorica`ListGraphs[n, k], g_ /; Combinatorica`ConnectedQ[g]], {k, n-1, n*(n-1)/2}]]]; row, {n, 1, 8}] // Flatten (* Jean-François Alcover, Jan 15 2015 *)

CROSSREFS    Cf. A008406, A054925.
Other versions of this triangle: A046751, A076263, A054923, A046742.
Row sums give A001349, column sums give A002905. A046751 is essentially the same triangle. A054923 and A046742 give same triangle but read by columns.
Main diagonal is A000055. Next diagonal is A001429. Largest entry in each row gives A001437.
Sequence in context: A326787 A246271 A049334 * A370167 A046751 A124478
Adjacent sequences: A054921 A054922 A054923 * A054925 A054926 A054927

KEYWORD    nonn,easy,nice,tabf

AUTHOR    N. J. A. Sloane

STATUS    approved

Figure 7: An example of one OEIS webpage. The identifier in this page is 'A054924'. This webpage includes the number sequence, number sequence offsets, number sequence references, number sequence links to other supplementary information, mathematical explanations, cross-references with other number sequences, and so on.

problem in Figure 10 as an example, if the guiding agent outputs 7 for the first example case, it matches the ground truth. If both the answers match the ground truth in example cases, we can say that the current generated problem is correct.

We ensure that the algorithmic problem takes the index of a sequence term as input and outputs the number corresponding to that index (if the input is 2, what we actually want is to output the 2-nd term of the sequence). Therefore, the code solution is the computational process of the general term.

We still manually select 5 to 7 items from the original sequence randomly as test cases, so that we can validate the correctness of the code solution. The test cases do not overlap with the example cases, and it is guaranteed that the second example case's next term in the sequence corresponds to the first test case (maintaining consistent bias).

Since our guiding agent is sufficiently powerful, it can correctly answer most of the constructed algorithmic problems, including the harder ones. As a result, more challenging problems are retained, keeping both the difficulty and diversity of our dataset.

In this way, we obtain a series of well-formatted and correct number sequence algorithmic problems, which we refer to as seed sequence data. We divide the sequences that are correctly constructed into two groups, preparing them separately for SFT and RL data construction.

**Why not use the same batch of data to construct both SFT and RL data?** This is because we aim to cover as many types of sequences (i.e., general term patterns) as possible.

**How can we ensure that the algorithmic problems are truly packaged?** (1) In the High-Density Sequence Data, the background introductions and mathematical descriptions of the current number sequence are already included. Therefore, the working agent can intuitively understand the mathematical logic of the sequence and, while ensuring the correctness, only needs to fabricate a story

1242
1243
1244
1245
1246
1247
1248
1249
1250
1251
1252
1253
1254
1255
1256
1257
1258
1259
1260
1261
1262
1263
1264
1265
1266
1267
1268
1269

> **The Prompt for Checking Enough Information**
>
> I will give you a sequence and all the relevant information about it.
> I would like to turn this sequence into an algorithmic problem about its general term formula.
> The problem must consist of the problem statement, the format requirements for the input and output, and two examples for input and output.
> Now, please first plan the steps required to generate an algorithm problem, and then evaluate whether the information I provided can meet the conditions for generating an algorithm problem by following those steps.
> Please output your response in JSON dictionary format: {"step": xxx, "step_judge": xxx, "is_able": xxx},
> where "step" represents the steps you planned, "step_judge" represents the thought process for each step's evaluation, and "is_able" indicates whether it is possible to generate an algorithm problem based on the provided information (True or False).
>
> <bos> (the sequence) <eos>
> [slot] (the relevant information ) [slot]

Figure 8: The prompt for the working agent to conduct self-planning on the problem generation and self-reflecting on whether each step contains enough information.

1270
1271
1272
1273
1274
1275
1276
1277
1278
1279

to wrap it. (2) For the generated example cases, we use human-written rules to determine whether they exist in the original number sequence and calculate the corresponding offsets. (3) To ensure the consistency between the example cases and the story descriptions, we have the guiding agent generate the output answers based on the input of the example cases and the problem description. We only retain the problems whose answers are generated correctly, thereby ensuring the consistency between all cases and the story description. (4) If we use the same agent that generates problem descriptions and solves the problems, this may lead to bias or limited diversity issues. So, we apply different agents. (5) To ensure the accuracy of the above processes, we do use powerful LLMs. Through these steps, we can rigorously ensure the alignment between the problem descriptions (stories) and all cases, as well as the correctness of all cases, and therefore no additional supervision is needed.

1280
1281

A.4.3 CASE-BASED REFLECTION INJECTION

1282
1283
1284

After obtaining the seed sequence data, we allow the working agent to directly generate the code solution for each algorithmic problem in the first group, as shown in Figure 12.

1285
1286

Since the problem description involves a story-based portrayal of the general term, the code solution represents the computational process for the general term of the sequence.

1287
1288
1289
1290
1291
1292
1293
1294
1295

**Sandbox and Unit Tests**    Imitating previous unit tests (Hui et al., 2024), we apply test cases to test the correctness of each code solution in an isolated sandbox environment. A sandbox environment for executing code (Cohn et al., 2024; Li et al., 2014) is a controlled and isolated setting where code can be run without affecting the host system or other applications. In this environment, the code is executed within a restricted space, preventing it from accessing sensitive resources, files, or system-level operations outside the sandbox. Sandboxes are commonly used for testing, experimentation, and security purposes, as they allow developers to execute potentially untrusted or experimental code safely. The goal is to mitigate risks, such as malware or unintentional system damage, by containing the code's actions and ensuring it can not interfere with critical parts of the system. Our code sets up a sandbox environment to safely execute user-provided Python code. It isolates the code by removing

> **The Prompt for Problem Generation**
>
> I will give you a sequence and all the relevant information about it.
> I would like to turn this sequence into an algorithmic problem about its general term formula.
> The problem must consist of the problem statement, the format requirements for the input and output, two example cases of input, output and their explanations (make it easier for problem solvers to understand).
> Please output your response in JSON dictionary format:
> {
> "thinking based on steps": xxx,
> "description": xxx,
> "input_format": xxx,
> "output_format": xxx,
> "example cases": [ {"input1":, "output1":, "explanation1":},
> {"input2":, "output2":, "explanation2":} ] ,
> }
>
> ##sequence## : <bos> (the sequence) <eos>
> ##relevant information## : [slot] (the relevant information )
> [slot]
> ##steps##: [slot](steps)[slot]

Figure 9: The prompt for algorithmic problem generation.

access to potentially dangerous built-in functions like open, exec, and eval, and replaces the print function with a safe version. We also redirect input and output to custom streams to capture them. The code is executed in a controlled environment with only a limited set of built-in functions available. If errors occur, they are caught and formatted with details, including the line number. Finally, we restore the system's original state after execution. This approach ensures safe, isolated execution of potentially risky code. Our sandbox code is adapted from the evaluation code of LiveCodeBench[6] (Jain et al., 2024) in the OpenCompass project (Contributors, 2023).

If a code solution fails on a test case, we ask the guiding agent to reflect and provide the reason for the failure (Figure 13). We then give that reason, the failed test case, along with the original code, back to the working agent to regenerate and correct the code solution. The prompt for the working agent to regenerate and correct the code is in Figure 14. Ultimately, through this continuous reflection process, we achieve a code solution that passes all the test cases. We inject each step of the case-driven reflection into the CoT. To ensure data diversity, we perform multiple samplings of both the problem descriptions and the initial attempt code solutions.

### A.4.4 SOLVABILITY-ESTIMATED SELECTION

**Some rationale for reward function design.** The CSSR reward function consists of a solvability term with an additional case-specific reward term.

A former study shows that if the additional reward exists in a weighted form, the theoretical optimal policy remains unchanged. On this basis, such multi-component rewards are common and do not disrupt the fundamental optimization objective.

---

[6]https://github.com/open-compass/opencompass/blob/main/opencompass/datasets/livecodebench/evaluator.py

> **description:**
> In the Kingdom of Numeria, the wise mathematician Elara has devised a unique festival where citizens create beautiful mosaic patterns using tiles of various sizes. However, to honor an ancient tradition, no tile in a mosaic can have a size that is a multiple of 3. The size of a tile is represented by a positive integer. Given a total area `n`, Elara challenges the citizens to determine how many distinct ways they can partition the area into tiles following the tradition. Each partition must consist of positive integers where none of the integers are multiples of 3, and the order of tiles does not matter (i.e., partitions are considered the same regardless of the order of tiles). Help the citizens of Numeria by writing a program that calculates the number of valid partitions for a given area `n`.Note: Two partitions are different if they have a different set of tile sizes, regardless of the order.
> **input_format:**
> The input consists of a single integer `n` (1 ≤ n ≤ 5000) representing the total area to be partitioned.
> **output_format:**
> Output a single integer representing the number of distinct valid partitions of `n` where no part is a multiple of 3.
> **example cases:**
> [ {"input1": 6, "output1": 7, "explanation1": For n=6, the valid partitions are:\\n[6], [5,1], [4,2], [4,1,1], [2,2,2], [2,2,1,1], [1,1,1,1,1,1]\\nThere are 7 valid partitions.}, ... ]
> **test cases:** ...

Figure 10: A generated example of a number sequence algorithmic problem.

In previous studies, the reward is constructed using historical success rates estimated from experience, and they also demonstrate that applying a log function can stabilize training. Therefore, in the first term, we apply a log-based shaping operation to the solvability component.

We train two models: `LLaMA3-8B-Instruct` and `Qwen2.5-7B-Instruct`. To more accurately estimate the difficulty of each problem, we chose `DeepSeek-R1-7B`, a model of comparable scale, to perform the rollouts. Since `DeepSeek-R1` is relatively newer and has stronger coding capabilities compared with the models to be trained (Fernandes et al., 2025), the difficulty scores it assigns tend to be slightly lower. However, this bias is unlikely to have a significant impact on the RL process (Qiao et al., 2025). To mitigate randomness, we set the number of rollouts $N$ to 32.

**Can we use the length of the CoT as an estimation of problem solvability?** When generating algorithmic problems with the prompt shown in Figure 9, we also have it output the CoT. Some previous studies (Chen et al., 2024c; Lei et al., 2024) show that when using LLMs to construct problems, the longer the CoT, the more difficult the generated problem tends to be. So, can we use the length of the CoT to represent the solvability of a problem? Or, in other words, is there a correlation between the length of the CoT and the solvability of the problem? We record the length of the CoT for each problem as well as the pass rate of the problem to calculate the correlation between the two in Figure 15. We normalize the thinking CoT into $[0, 1]$ and compute the Pearson correlation coefficient (Cohen et al., 2009) and Spearman correlation coefficient (Hauke & Kossowski, 2011) between the two variables. The values of the two are $-0.10$ and $-0.21$, respectively, and it can be seen that the correlation between the two is not significant in this experiment. Therefore, we do not use the length of the thinking CoT to represent the solvability of a problem.

> **The Prompt for Example Cases' Outputs Generation**
>
> I will now give you an algorithmic problem along with two input examples (numbers).
> Please directly provide the corresponding answers (numbers) for these two inputs.
> Please output your response in JSON dictionary format:
> {"reason1": xxx, "answer1": xxx, "reason2": xxx, "answer2": xxx},
> where "reason1" and "reason2" represent your thought process for the two input examples, and "answer1" and "answer2" are your answers (please provide the numbers directly, with no extra output).
>
> ## problem description## : [slot] (problem description) [slot]
> ## input1## : [slot] (input1) [slot]
> ## input2## : [slot] (input2) [slot]

Figure 11: The prompt for the guiding agent directly outputs the results so that we can determine whether the current problem is correct.

> **The Prompt for Code Solution Generation**
>
> I will now give you an algorithmic problem.
> Please give me your code solution with Python.
> Please respond in JSON dictionary format: {"thought": xxx, "code": xxx},
> where the "thought" section represents your reasoning process for the problem, and the "code" section should directly give a python code.
>
> ##problem description## : [slot] (problem description) [slot]
> ##input format## : [slot] (input format) [slot]
> ##output format## : [slot] (output format) [slot]
> ##example1## : [slot] (example1) [slot]
> ##example2## : [slot] (example2) [slot]

Figure 12: The prompt for the code solution generation.

### A.5 SUPPLEMENTARY INFORMATION OF THE *CodeSeq* DATASET

Based on the above process, we construct SFT and RL data with number sequences, then form a post-training dataset *CodeSeq*.

A standard SFT input format in *CodeSeq* is shown in Figure 16, and a standard SFT output format in *CodeSeq* is shown in Figure 17. As other powerful reasoning models, we use the CoT technique (Yang et al., 2024b) to guide the model's deep reasoning process. In the output format, we store the CoT field and the final answer field separately. As for RL data, we use the same 'input' as in Figure 16, but in the output, we only retain the 'code' field.

**The Prompt for Giving Reasons**

I will now give you an algorithmic problem, its python code and one case.
Please tell me why the case works and why the code fails on the case.
Please output your response in JSON dictionary format:
{"work_reason": xxx, "failed_reason": xxx},
where "work_reason" and "failed_reason" represent why the case works and why the code fails on the case.

##problem description## : [slot] (problem description) [slot]
##input format## : [slot] (input format) [slot]
##output format## : [slot] (output format) [slot]
##example1## : [slot] (example1) [slot]
##example2## : [slot] (example2) [slot]
##python code## : [slot] (code) [slot]
##failed case input## : [slot] (failed case input) [slot]
##failed case output## : [slot] (failed case output) [slot]

Figure 13: This prompt is inputted into the guiding agent to generate the reason why such a case fails on the current code solution.

**The Prompt for Code Solution Updation**

I will now give you an algorithmic problem, the code solution for this problem and a test case that the code fails.
Please give me your updated code solution with Python.
Please respond in JSON dictionary format: {"thought": xxx, "code": xxx},
where the "thought" section represents your reasoning process for the problem, and the "code" section should directly give a python code.

##problem description## : [slot] (problem description) [slot]
##input format## : [slot] (input format) [slot]
##output format## : [slot] (output format) [slot]
##example1## : [slot] (example1) [slot]
##example2## : [slot] (example2) [slot]

##origin code## : [slot] (origin code) [slot]
#case input## : [slot] (case input) [slot]
#case output## : [slot] (case input) [slot]
#work reason## : [slot] (work reason) [slot]
#failed reason## : [slot] (failed reason) [slot]

Figure 14: The prompt for the working agent to regenerate and correct the code.

It is worth noting that our SFT data includes a small number of short training samples (e.g., cases where a single line of 'print' suffices to express the general formula). This is intended to increase the diversity of the synthetic data, helping the model learn problem-solving primitives. There aren't

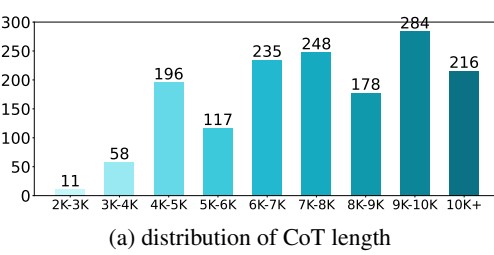 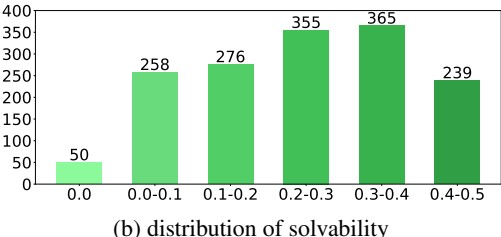

| (a) distribution of CoT length | (b) distribution of solvability |

Figure 15: Exploring the relationship between the length of the thinking CoT and the solvability.

many such short training samples, and this is intended to avoid bias and prevent issues such as lazy planning (Pasricha & Roncone, 2024) (To make the model believe that shorter outputs are better).

---

**input:**

You are now an algorithm competition participant. Based on the following algorithm problem, you need to provide the problem-solving approach and the Python code.

Here is the problem statement: **[slot] (problem description) [slot]**
input format for this problem is: **[slot] (input format) [slot]**
output format for this problem is: **[slot] (output format) [slot]**

I will give you two examples to help you better understand the problem and come up with the solution approach and Python code.

### example 1
input1: **[slot] (input1) [slot]**
output1: **[slot] (output1) [slot]**
The explanation for this example is: **[slot] (explanation1) [slot]**

### example 2
input2: **[slot] (input2) [slot]**
output2: **[slot] (output2) [slot]**
The explanation for this example is: **[slot] (explanation2) [slot]**

When you output your solution approach, you can write an initial solution code based on your first thoughts.Then, please create examples and test them on your own. Afterward, reflect and make corrections to your code based on the examples.

Now, please provide the Python solution approach and code for this problem in {\"cot\":xxx,\"code\":xxx} dictionary format, where \"cot\" represents your solution approach, and \"code\" represents the final answer code.

---

Figure 16: The standard input of one SFT training data in *CodeSeq*.

**Why 0.46?** The value $0.46$ represents an approximation of $15/32$. We chose this number primarily to keep the amount of RL training data around $1.5K$ and ensure the data quality. On one hand, we show that training with easy samples is less effective (Figure 3 (b)), so we prioritize selecting difficult samples (with solvability less than $0.5$) for the model to learn from. In contrast, if we use fewer data samples (such as less than $1.2K$), our prior experiments and the results in Figure 3 ((b) only hard problems) demonstrate that it is difficult to achieve optimal performance. On the other hand, the

```
output:
{
"cot":
"The Python code for this algorithm problem, based on my initial
thoughts, is:
    [slot](code1)[slot]
The solution code above did not pass the following test case:
    [slot](input1)[slot] [slot](output1)[slot]
The reason this test case is valid is:
    [slot](valid rason1)[slot]
The reason the code fails for this case is:
    [slot](failed reason1)[slot]
Based on this test case, the revised solution code is:
    [slot](code2)[slot]
...
This code passes all the test cases, so it is the correct solution.",

"code":
    [slot](answer code)[slot]
}
```

Figure 17: The standard output of one SFT training data in *CodeSeq*.

total number of samples with solvability less than $0.5$ is around $1.7$K. If all $1.7$K data are used for training, the results on GTG are approximately $41.89$ on `LLaMA3-8B-Instruct`, which is lower than that of *CodeSeq*$_{\text{RL}}$. In summary, for the choice of the solvability upper bound, our constraints are: (1) filtering out simple training samples, so it needs to be less than $0.5$; (2) the number of training samples is at least $1.2$K (since performance keeps improving when the sample size is less than $1.2$K); (3) choosing $0.5$ yields $1.7$K training samples, but the performance drops. Therefore, we chose $0.46$ as the threshold, corresponding to $1.5$K training samples, for RL.

**Data Statistics**  After going through the strict and complex data generation process shown in Figure 2, we perform statistical analysis for each step in Table 5. For each algorithmic problem derived from a number sequence, if it fails at any step in the pipeline, we discard that data. In the end, we retained 20K valid algorithmic problems for generating SFT and RL data. As shown in the table, our final post-training synthetic data undergoes rigorous filtering and is of relatively high quality.

| | |
|---|---|
| # scapped number sequence | 270K |
| # high-density number sequence | 100K |
| # validated problems | 20K |
| # seed sequence data #1 | 10K |
| # seed sequence data #2 | 10K |
| # passed SFT data | 2K |
| # resampled SFT data | 5.5K |
| # passed RL data | 4K |
| # seleted RL data | 1.5K |

Table 5: The sample statistics at each stage in the synthetic data pipeline. The data presented in this table are not the accurate values.

**Pricing**  When constructing the synthetic data, we primarily use the API Keys of `DeepSeek-V3` and `o3-mini-high` for our experiments. The total cost is approximately USD $2,500$ and RMB $400$. We invest a significant amount of funding in trial-and-error and data synthesis, which helps ensure the quality and effectiveness of the data. The expenses mentioned above are not solely for

data synthesis. A small part of them is also included in the evaluation, which we will explain in detail in the experimental section.

### A.6 DETAILS FOR TRAINING AND EVALUATION

#### A.6.1 LLM BACKBONES

We conduct SFT and RL experiments on two widely used LLMs: `LLaMA3-8B-Instruct` and `Qwen2.5-7B-Instruct` with our post-training dataset *CodeSeq*. We do not choose stronger models such as `Qwen3` or `LLaMA3 +` for two reasons. First, the `Qwen3` open-source models do not cover a wide range of parameter sizes, and many of the currently available models (e.g., the distilled version of `DeepSeek-R1`) are derived from the `Qwen2.5` series. So `Qwen-2.5` is used more widely. In particular, we later conduct experiments with a dense 32B model, which `Qwen3` does not have such a parameter size. Overall, we opt to continue using `Qwen2.5`. Second, the `LLaMA3 +` models and `Qwen2.5` are released nearly around the same time. The difference between the two is not as large as enough. To better demonstrate the robustness of our data, we chose to use the original `LLaMA3` model, which is more distinct.

**LLaMA3-8B-Instruct** (Grattafiori et al., 2024) `LLaMA3-8B` is an advanced LLM developed by Meta, featuring 8 billion parameters. It is part of the Llama 3 family. This model is built on an optimized Transformer architecture and trained on a diverse dataset of over 15 trillion tokens. The training dataset includes a significant amount of code and covers over 30 languages, with more than 5% of the data being non-English. `LLaMA3-8B` is particularly designed to excel in instruction-based tasks, making it highly effective for scenarios requiring precise and context-aware responses.

**Qwen2.5-7B-Instruct** (Qwen et al., 2025) `Qwen2.5-7B` is a powerful LLM developed by Alibaba's ModelScope team, featuring 7.6 billion parameters. It is designed to excel in various NLP tasks, with notable strengths in long-context understanding, multilingual support, and specialized capabilities for coding and mathematical tasks. This model supports up to 128K tokens for context understanding and can generate up to 8K tokens of text, making it highly effective for long-text generation and structured data processing. What's more, `Qwen2.5-7B` is trained on 18T data.

#### A.6.2 MIX TRAINING DURING SFT

To maintain the models' instruction-following and other inherent abilities when conducting SFT, we mix *CodeSeq*$_\text{SFT}$ with the latest post-training corpus Tülu 3.

**Tülu 3** is a comprehensive dataset developed by the Allen Institute to advance the post-training of LLMs. The Tülu 3 dataset is designed to enhance language models' performance through SFT and RL. It includes a mixture of data from various sources, covering a wide range of natural language processing tasks such as instruction following, mathematical reasoning, and code generation.

The training data of Tülu 3 comes from various sources, like the instruction dataset: FLAN v2, OpenAssistant, WildChat, and No-Robots; the math instruction dataset: NuminaMath, SciRIFF, and OpenMathInstruct; also, some code instruction datasets: CodeAlpaca; the rest data is from human-made instruction data with `GPT-4o` and `Claude3`. So, we ensure that it does not conclude any data from the benchmarks.

During the training process, we remove samples longer than 3K tokens and exclude all training samples related to code (since we focus on code problems) with `Qwen2.5-7B-Instruct` filtering such code-area data. Finally, we retain over 800K training samples of Tülu 3. It is worth noting that we only perform around $1,000$ steps of SFT, so not all of the 800K Tülu 3 data is trained.

To improve the models' reasoning ability while maintaining their other capabilities, particularly instruction-following ability, we calculate the average number of tokens in the Tülu 3 and *CodeSeq* datasets. We assign a weight ratio of 5:1 to these two datasets for mixed training.

### A.6.3 TRAINING IMPLEMENTATIONS AND PARAMETERS

We conduct both SFT and RL on two widely used LLMs. We conduct SFT and RL based on the InternTrainer[7] framework and VERL[8] (Sheng et al., 2024) framework with 8 NVIDIA-L20Y separately. We not only compare the performance differences of the models before and after training, but also include `InternLM2.5` and `DeepSeek-R1`, each coming in two parameter sizes, as additional baselines. The training parameters of SFT and RL are shown in Table 6 and Table 7, respectively. The parameter $\lambda$ in Formula 2 is set to 0.9.

| | |
|---|---|
| total-steps | $1,000$ |
| epochs | 1 |
| bsz | 16 |
| gradient-accumulation | 16 |
| micro-bsz | 1 |
| seq-len | $5,120$ |
| max-length-per-sample | $5,120$ |
| min-length | 50 |
| num-worker | 4 |
| loss-label-smooth | 0 |
| lr | $1e{-}6$ |
| warmup-ratio | 0.1 |
| weight-decay | 0.01 |
| adam-beta1 | 0.9 |
| adam-beta2 | 0.95 |
| adam-eps | $1e-8$ |
| fp16-initial-scale | $2**14$ |
| fp16-min-scale | 1 |
| fp16-growth-interval | $1,000$ |
| fp16-growth-factor | 2 |
| fp16-backoff-factor | 0.5 |
| fp16-max-scale | $2**24$ |
| zero1-size | 8 |
| tensor-size | 1 |
| pipeline-size | 1 |
| weight-size | 1 |

Table 6: The training parameters of SFT.

| | |
|---|---|
| total-steps | 300 |
| bsz | 64 |
| actor.ppo-mini-bsz | 64 |
| promot-max-length | $2,048$ |
| response-max-length | $2,048$ |
| lr | $1e{-}6$ |
| rollout.n | 8 |
| warmup-step | 0.1 |
| warmup-style | cosine |
| actor.ulysses-sequence-parallel-size | 1 |
| actor.use-kl-loss | True |
| actor.entropy-coeff | 0.0 |
| actor.kl-loss-coef | 0.0 |
| actor.ppo-micro-bsz-per-gpu | 4 |
| ref.log-prob-micro-bsz-per-gpu | 64 |
| rollout.log-prob-micro-bsz-per-gpu | 64 |
| rollout.tensor-model-parallel-size | 1 |

Table 7: The training parameters of RL.

### A.6.4 BENCHMARKS

We test the tuned models on three code benchmarks: Humaneval (Chen et al., 2021), MBPP (Austin et al., 2021) and LiveCodeBench (Jain et al., 2024), along with three comprehensive reasoning benchmarks: MMLU (Hendrycks et al., 2021), BBH (Suzgun et al., 2022), and GaoKaoBench (Zhang et al., 2024b). Next, we will introduce these benchmarks one by one.

**Humaneval** consists of 164 hand-crafted programming challenges that are comparable to simple software interview questions, each with a function signature, natural language description, and unit tests to validate the correctness of generated code.

---

[7]https://github.com/interntrainer
[8]https://github.com/volcengine/verl

**MBPP** (Mostly Basic Python Problems) consists of around $1,000$ crowd-sourced Python programming problems, each with a task description, code solution, and three automated test cases.

**LiveCodeBench** is a comprehensive benchmark designed to evaluate the coding capabilities of LLMs through diverse and challenging programming tasks. It focuses on real-world scenarios, testing code generation, debugging, and optimization to advance AI-driven software development.

**MMLU** The MMLU (Massive Multitask Language Understanding) benchmark is a comprehensive evaluation tool designed to assess the knowledge and reasoning capabilities of LLMs across a wide range of academic and real-world subjects.

**BBH** The Big Bench Hard (BBH) benchmark is a collection of challenging tasks designed to evaluate the reasoning and logical abilities of LLMs.

**GaoKaoBench** The GaoKaoBench is an evaluation framework that uses Chinese college entrance examination (Gaokao) questions as its dataset to assess the language understanding and logical reasoning capabilities of LLMs. It includes a comprehensive collection of questions from 2010 to 2023. For convenience in evaluation, we select only objective questions for testing.

A.6.5 COMPARED MODELS

In Table 2, we compare our trained models with many baselines. In this part, we will introduce these baselines. At the same time, the models used in this paper will also be presented here.

**o3** `o3`[9] is a next-generation AI platform engineered to redefine intelligent automation and decision-making. By integrating SOTA machine learning with domain-specific expertise, `o3` delivers actionable insights, optimizes complex workflows, and adapts dynamically to evolving challenges. Designed for scalability, it serves industries ranging from healthcare to finance.

**Qwen2.5-32B-Instruct** `Qwen2.5-32B`[10] is Alibaba's advanced 32-billion-parameter AI model that delivers exceptional performance in multilingual understanding, complex reasoning, and coding tasks. The model demonstrates competitive benchmark scores, particularly in Chinese-language applications, while maintaining strong English capabilities, making it ideal for enterprise solutions and research applications across diverse linguistic contexts.

**InternLM2.5-7B-chat** `InternLM2.5-7B`[11] (Cai et al., 2024b) is a powerful yet compact 7-billion-parameter language model developed by Shanghai AI Laboratory, offering exceptional performance in Chinese and English tasks while maintaining efficient deployment capabilities. This latest version demonstrates significant improvements in mathematical reasoning and coding tasks compared to its predecessors. Designed for cost-sensitive applications, it delivers near-70B-model performance at a fraction of the computational costs.

**InternLM2.5-20B-chat** `InternLM2.5-20B`[12] is Shanghai AI Laboratory's mid-sized powerhouse, delivering 20-billion-parameter performance that rivals many larger models. This iteration shows dramatic improvements in Chinese-English bilingual tasks while maintaining exceptional cost-efficiency, featuring a 32K-token context window and enhanced reasoning capabilities.

**DeepSeek-R1-7B** `DeepSeek-R1-Distill-Qwen-7B`[13] is a distilled version of DeepSeek's 7-billion-parameter language model on `Qwen2.5-Math-7B`, optimized for efficient deployment while retaining strong performance in reasoning and coding tasks. With enhanced inference speed and reduced computational requirements, it delivers competitive accuracy in various benchmarks.

---

[9]https://openai.com/index/introducing-o3-and-o4-mini/
[10]https://huggingface.co/Qwen/Qwen2.5-32B
[11]https://huggingface.co/internlm/internlm2_5-7b-chat
[12]https://huggingface.co/internlm/internlm2_5-20b-chat
[13]https://huggingface.co/deepseek-ai/DeepSeek-R1-Distill-Qwen-7B

**DeepSeek-R1-32B** `DeepSeek-R1-Distill-Qwen-32B`[14] is a distilled version of the 32-billion-parameter model on `Qwen2.5-32B`, optimized for efficiency while maintaining robust performance in complex reasoning and multilingual tasks. This compact yet powerful variant achieves near-original model accuracy with significantly reduced computational costs, making it ideal for scalable enterprise deployment.

**DeepSeek-R1** `DeepSeek-R1`[15] is a SOTA open-source large language model developed by Chinese AI startup DeepSeek. It excels in complex tasks such as mathematical problem-solving, coding, and logical reasoning, achieving performance comparable to OpenAI's `o1` model. `DeepSeek-R1` employs a Mixture-of-Experts (MoE) architecture with a total of 671 billion parameters, of which 37 billion are activated per token during inference, balancing performance and computational efficiency.

**DeepSeek-V3** `DeepSeek-V3`[16] is a cutting-edge open-source LLM developed by the Chinese AI company DeepSeek. It features a Mixture-of-Experts (MoE) architecture with a total of 671 billion parameters, activating 37 billion per token during inference, balancing performance and efficiency. Trained on 14.8 trillion high-quality tokens, `DeepSeek-V3` demonstrates strong capabilities in reasoning, coding, and multilingual tasks.

**Claude-Sonnet-4** `Claude-Sonnet-4`[17] is a hybrid reasoning model from Anthropic that builds on Sonnet 3.7, offering improvements especially in coding, instruction-following, and reasoning. It features two modes: near-instant response for when speed is important, and extended thinking for deeper, multi-step reasoning. Compared to its predecessor, it is better at following nuanced instructions, reducing errors, navigating complex codebases, and delivering more reliable outputs.

### A.6.6 OPENCOMPASS

We employ OpenCompass[18] (Contributors, 2023), which is an LLM evaluation platform, supporting a wide range of models, to evaluate the results. It features a wide range of capabilities, including language understanding, reasoning, coding, and long-text generation, and provides a fair and reproducible benchmark for model evaluation. For the GTG task, we chose `gen pass@1` (Chen et al., 2021) (where $n = 32$) as the evaluation metric. In code reasoning: Humaneval and LiveCodeBench use `gen pass@1`, while MBPP utilizes `accuracy`. For the three comprehensive reasoning tasks: MMLU, BBH, and GaoKaoBench, we apply `native average`, `weighted average`, and `accuracy` as evaluation metrics, respectively.

Although our evaluation results may differ somewhat from the original benchmark results, for the same benchmark, we use the same evaluation framework and settings, so the comparison is fair.

### A.6.7 EXPLANATIONS ON LLMS WITH AN ENORMOUS NUMBER OF PARAMETERS

For all open-source models, we perform local deployment and conduct inference tests. For larger models such as `DeepSeek-R1` and `DeepSeek-V3`, we deploy them on 16 80GB GPUs and perform inference using SGLang[19] (Zheng et al., 2024). For closed-source models, we access them via API calls. We apply the same settings for all the models in Table 8 for GTG tasks.

### A.6.8 ANALYSE ON FAILED CASES

We conduct a statistical analysis and investigation on the cases where the Qwen2.5 model fails both before and after training.

---

[14]https://huggingface.co/deepseek-ai/DeepSeek-R1-Distill-Qwen-32B
[15]https://huggingface.co/deepseek-ai/DeepSeek-R1
[16]https://huggingface.co/deepseek-ai/DeepSeek-V3
[17]https://www.anthropic.com/news/claude-4
[18]https://github.com/open-compass/opencompass
[19]https://github.com/sgl-project/sglang

| | |
|---|---|
| rollouts | 32 |
| temperature | $min(max\_temp, 1.0)$ |
| max-response-length | $min(max\_len, 10 * 1024)$ |
| rolling-timeout | 10 |

Table 8: The settings for evaluating the GTG task on various models.

We find that the majority of failures (about 60%) are due to the general term formulas of the number sequences involving compositional rules. For example, the sequence 10, 3, 2, 2, 3, 3, 3, . . . , is generated by a formula that combines square root operations, recursion, and modulo operations.

Another portion (about 30%) of the number sequences contains hidden variables, where the sequence alone is insufficient to infer the rule, requiring reconstruction of the hidden state. For instance, the sequence 1, 1, 2, 4, 3, 7, 5, 12, 7, . . . actually represents a robot moving in the directions right, down, left, and up in order, with each step's displacement equal to the sum of the previous two steps. The general term of the sequence corresponds to the robot's Manhattan distance (the hidden variable in this case) from the starting point.

## A.7 SUPPLEMENTARY INFORMATION ON THE EXPERIMENTS

### A.7.1 EXPLANATIONS ON MAIN RESULTS

The term "domain" in Table 2 refers to number sequence inductive reasoning tasks. 'in-domain' refers to the GTG task, which involves calculating the general term of a sequence using code. Since the sequence is presented in the form of code, we consider code reasoning as a close domain. Other reasoning tasks can be thought of as OOD.

In the main experiments, we use the same configuration for each benchmark to ensure fairness. For all public benchmarks except the GTG task, we adopt the default evaluation settings provided by OpenCompass. For the GTG task, we apply the same settings for all the models in Table 8.

We do not conduct OOD testing for the baseline models or the ablation studies, as our OOD benchmarks are intended solely to verify that *CodeSeq* does not compromise the general capabilities of the backbones, rather than to demonstrate improvements in general performance. Moreover, both DeepSeek-R1-7B and DeepSeek-R1-32B fail to produce valid predictions on the MBPP benchmark when evaluated using the same OpenCompass configuration. Therefore, to ensure fairness, we do not report their results.

### A.7.2 TRAINING WITH LARGER MODELS

To further demonstrate the effectiveness of our data, we conduct the same training as in the main experiments on a larger model Qwen2.5-32B-Instruct. The results in Table 9 show that our training data are effective for models of various sizes.

| | in-domain | close-domain | | | out-of-domain | | |
|---|---|---|---|---|---|---|---|
| | GTG | Humaneval | MBPP | LCBench | MMLU | BBH | GaoKao |
| Qwen2.5-32B | 38.04 | 89.02 | 83.27 | 59.51 | 84.26 | 84.42 | 90.48 |
| w/ *CodeSeq*SFT | 39.82 | 90.52 | 85.33 | 61.32 | 83.89 | 84.72 | 90.52 |
| w/ *CodeSeq*RL | 68.26 | 92.03 | 84.54 | 62.36 | 83.77 | 84.95 | 90.85 |
| w/ *CodeSeq* | 75.23 | 92.37 | 85.59 | 62.60 | 84.01 | 85.05 | 90.87 |

Table 9: Results on training with a larger model Qwen2.5-32B-Instruct.

### A.7.3 TRAINED MODELS FOR NEXT NUMBER PREDICTION

We mention the next number prediction task above. Now, we also test our trained models LLaMA3-8B-Instruct and Qwen2.5-7B-Instruct on this task. Results are in Table 10.

The experimental results show that our training data are also effective for this task. It is worth noting that our results are relatively lower compared to the Table 2. This is mainly because different prompts and number sequence terms are employed for testing across tasks, and in the main experiments, we performed 32 rollouts to compute the average results.

| Model | next number prediction |
|---|---|
| LLaMA3-8B | 8.0 |
| w/ *CodeSeq* | 33.0 |
| Qwen2.5-7B | 17.0 |
| w/ *CodeSeq* | 38.0 |

Table 10: Results of trained models on the next number prediction task.

### A.7.4 EXPLANATIONS ON DIFFERENT TRAINING SETTINGS

**Different CoT** When using different types of CoT for SFT training, we ensure that the number of training data is kept consistent. Figure 18 and Figure 19 demonstrate the CoT example for 'CaseEx' and 'NL', meaning providing cases along with explanations in the CoT to enhance the models' ability to generate cases and using natural language explanations instead of providing failed cases during the reflection process, separately.

**output:**
{
"cot":
"

   [slot](one term in the number sequence)[slot]
   [slot](the explannation of the term)[slot]

   [slot](one term in the number sequence)[slot]
   [slot](the explannation of the term)[slot]

......
   [slot](one term in the number sequence)[slot]
   [slot](the explannation of the term)[slot]
",

**"code":**
   [slot](answer code)[slot]
}

Figure 18: An example of the 'CaseEx' CoT.

**Different Sovability** Under the four settings, we ensure that the number of training samples is fixed at 1K. For 'HighSov', we randomly select samples with solvability between $0.7$ and $1.0$; for 'LowSov', we sample instances with solvability between $0.0$ and $0.3$; for 'Random', we randomly choose samples from the entire solvability range; and for 'CodeSeq', we sample instances with solvability between $0.0$ and $0.46$.

**Different Reward Functions** Below, we sequentially present the three reward functions we compare: 'Binary', 'PassRate', and 'NoLog'.

$$\mathcal{R}_{\text{Binary}} = \begin{cases} 1 & \text{if all cases pass} \\ 0 & \text{else} \end{cases} \tag{3}$$

```
output:
{
"cot":
"The Python code for this algorithm problem, based on my initial
thoughts, is:
    [slot](code1)[slot]

I don't think this code is correct, because: xxxxx
So, the revised solution code is:
    [slot](code2)[slot]

I don't think this code is correct, because: xxxxx
So, the revised solution code is:
    [slot](code3)[slot]
...

I think this is the correct solution.",

"code":
    [slot](answer code)[slot]
}
```

Figure 19: An example of the 'NL' CoT.

$$\mathcal{R}_{\text{PassRate}} = 1 - \hat{Sov} \tag{4}$$

$$\mathcal{R}_{\text{NoLog}} = \begin{cases} -1, & \text{if formatted error} \\ 0, & \text{if any test case fail to execute} \\ \lambda(1 - \hat{Sov}) + (1-\lambda)\frac{N_{tc}}{N_c} & \text{if all test cases pass} \end{cases} \tag{5}$$

Experimental results show that using only the pass rate leads to a volatile RL training process. Although a binary reward function can increase the final reward, the improvement is relatively slow because it does not provide fine-grained differentiation between different code solutions.

**Whether the reward we design would cause reward design flaws?** There is a viewpoint that our designed reward function has a bias. If the sequence is long, then the polynomial that fits it is essentially unique. In this case, there will only be one valid pattern. However, if the sequence is short and we allow higher-degree polynomials, then many different patterns can all fit the given numbers. If the reward is based on the number of valid patterns, it will naturally favor situations where multiple patterns exist. That means the model is encouraged to get better at handling cases that are inherently ambiguous, since many different polynomials can explain them. At the same time, it becomes worse at handling cases where the sequence is longer and the polynomial is uniquely determined. In short, the reward design pushes the model toward ambiguous patterns rather than guiding it toward the correct and more meaningful ones.

We can approach this question from two perspectives. (1) Most of our GTG samples do not involve polynomials. One of the reasons we package sequences into algorithmic problems is that many of the underlying math rules cannot be intuitively expressed using mathematical formulas (such as polynomials), as in the introduction. We understand the bias described above, but the fact remains that most of our underlying formulas are indeed not polynomial-based. (2) If a number sequence has multiple patterns that satisfy it during rollout, we can consider that the sequence has multiple code solutions. Therefore, for the LLM currently attempting the problem, it is a relatively easy task. One characteristic of inductive reasoning is that a unique answer does not necessarily exist, so this is not an issue. Consequently, in Formula 2, we assign a lower reward to this type of problem.

|  | in-domain | close-domain | | | out-of-domain | | |
|---|---|---|---|---|---|---|---|
| **Models** | GTG | Heval | MBPP | LCBench | MMLU | BBH | GaoKao |
| `Qwen2.5-7B` | 26.89 | 82.31 | 71.59 | 41.97 | 75.49 | 64.80 | 75.75 |
| w/ GRPO | 63.36 | 83.73 | 73.51 | 42.33 | 75.12 | 64.89 | 76.81 |
| w/ DAPO | 62.22 | 82.31 | 71.80 | 42.06 | 76.53 | 64.22 | 75.70 |
| w/ PPO | - | - | - | - | - | - | - |
| w/ DPO | 27.90 | 81.44 | 74.71 | 42.21 | 76.32 | 62.11 | 78.44 |

Table 11: Results on training with different RL algorithms on `Qwen2.5-7B-Instruct`.

| | | | |
|---|---|---|---|
| SFT | **whole time** | **time per** 100 **step** | **weight convertion time** |
| | $7,860s$ | $786s$ | $30s$ |
| RL | **whole time** | **val time portion** | **weight convertion time** |
| | $17,124s$ | $0.2$ | $70s$ |

Table 12: The training time for both the SFT and RL stages with `LLaMA3-8B-Instruct`.

### A.7.5 DIFFERENT RL METHODS

We primarily use the GRPO algorithm to train our RL data. Meanwhile, we also compare it with other RL algorithms, DAPO (Yu et al., 2025), PPO (Schulman et al., 2017) and DPO (Rafailov et al., 2024), to draw more convincing conclusions on `Qwen2.5-7B-Instruct`.

As shown in Table 11, the PPO algorithm leads to model collapse, while the DPO algorithm, despite achieving surprisingly strong OOD performance on certain benchmarks, performs worse overall compared to GRPO. For the DAPO algorithm, although the model can maintain OOD performance, its performance on the GTG task and the three code benchmarks is inferior to GRPO. Therefore, we ultimately chose the GRPO algorithm.

### A.7.6 THE TRAINING TIME

We also record the training time for both the SFT and RL stages with `LLaMA3-8B-Instruct`. As shown in the Table 12, although our model achieves significant improvements in both inductive reasoning and code reasoning, it requires no more than 5 hours of training, whether in SFT or RL, to reach optimal performance. This clearly expresses the effectiveness of our data.

### A.7.7 EXPLANATIONS ON SCALING BEHAVIORS

**Reflection Round Scaling** We also ensure consistency in the number of SFT training samples. Specifically, within the final 5.5K SFT training dataset, we count the number of samples corresponding to different numbers of reflection rounds: $num_0$, $num_1$, $num_2$, $num_3$, $num_4$, and $num_5$. We then select the minimum among these as the SFT data size for our experiment. The statistics are shown in Table 13, and the final selected number is 630.

**Number Sequence Pattern Scaling** In this section, we also randomly select 300, 600, 900, and $1,200$ training samples from *CodeSeq* for our experiments. We make sure that each part of these samples is utilized for both models' SFT and RL.

### A.7.8 EXPLANATIONS ON REASONING WITH CASES

We prompt two LLMs to make cases themselves while producing code solutions, to verify the correctness of the code. This approach allows us to assess whether *CodeSeq* enables the model to learn how to generate correct cases. We deploy the prompt shown in Figure 16. We force the LLMs to generate at least one case, and if all of the cases exist in the current sequence, we consider it to be correct. If the LLMs fail to generate at least one case, it is prompted to regenerate until the output satisfies our requirements.

| | |
|---|---|
| $num_0$ | 1,045 |
| $num_1$ | 1,151 |
| $num_2$ | 742 |
| $num_3$ | 630 |
| $num_4$ | 1,110 |
| $num_5$ | 819 |

Table 13: The number of samples corresponding to different numbers of reflection rounds among the SFT data.

### A.7.9  THE ADVANTAGES OF OUR TRAINING PROCESS

We compare the advantages of our training process with a recent, widely recognized work. Wang et al. (2024c) is just similar to our SFT data construction process, but their approach is more dependent on human intervention. Meanwhile, their evaluation task is limited, and they only use non-trainable models (e.g., GPT-4). Our synthetic data not only includes SFT and RL data but also validates the performance of seven sub-tasks across different domains on trainable open-source models. More importantly, the core of our work lies in ensuring the fully automated construction of training data that incorporates the correct reasoning process, with the goal of enhancing the fundamental inductive reasoning abilities of LLMs—rather than merely prompting LLMs. We list the differences in Table 14.

| Data Source | (Wang et al., 2024c) | Ours |
|---|---|---|
| Human Intervention | Yes | No |
| Model Trainability | No | Yes |
| Multi-domain Validation | No | Yes |
| Practicality Validation | No | Yes |

Table 14: Comparison between one previous work and our approach.

### A.7.10  THE OOD INDUCTIVE TASKS

To evaluate the effectiveness of our data on various inductive reasoning tasks, we evaluate Qwen2.5-7B on four OOD inductive reasoning benchmarks. The results in Table 15 show that after being trained on our data, the model exhibits clear transferability to these tasks, indicating that its intrinsic inductive reasoning capability has been improved.

| | ARC | Listfunctions | CodeARC | Simbo |
|---|---|---|---|---|
| Qwen2.5-7B | 0.0 | 25.0 | 29.2 | 39.0 |
| w/ CodeSeq | 0.9 | 28.0 | 30.1 | 42.0 |

Table 15: Qwen2.5-7B on four OOD inductive reasoning benchmarks.

Since we use different metrics and evaluation methods in the tables, it is not straightforward to directly compare the difficulty of ARC and other tasks with our GTG task. We now re-evaluate the GTG task using two strong closed-source reasoning models with the same evaluation settings as ARC. The results in Table 16 indicate that our GTG task is more challenging compared to existing inductive reasoning tasks.

### A.7.11  EXPERIMENTS ON THE DEDICATED REASONING MODEL

| Models | GTG | Arc | ListFunctions |
|---|---|---|---|
| o3-mini-high | 0.28 | 0.35 | 0.55 |
| Claude-Sonnet-4 | 0.21 | 0.24 | 0.52 |

Table 16: Performance of different models on three tasks in the same settings.

We also employ a dedicated reasoning model Qwen2.5-7B-Coder as the base model for training, and the experimental results in Table 17 demonstrate that our data remains effective on such reasoning models.

| qwen2.5-coder-7B | GTG | humaneval | mbpp | lcb | mmlu | bbh | gaokao |
|---|---|---|---|---|---|---|---|
| vanilla | 30.11 | 88.41 | 78.99 | 44.57 | 69.91 | 67.38 | 67.54 |
| w/ CodeSeq | 70.05 | 89.93 | 79.55 | 45.60 | 69.82 | 67.11 | 67.58 |

Table 17: Evaluation results of Qwen2.5-coder-7B with and without CodeSeq.

### A.7.12 SIGNIFICANCE OF THE TEST SET

To demonstrate the representativeness of our test set, we expand the test set to the same scale as the training data (1,000). The results in Table 18 show that the performance difference between the two scales is, on average, less than 0.57 points, which is relatively small.

| Model | GTG(200) | GTG(1000) |
|---|---|---|
| LLaMA3-8B | 9.27 | 8.30 |
| w/ CodeSeq | 44.22 | 44.89 |
| Qwen2.5-7B | 26.89 | 27.15 |
| w/ CodeSeq | 69.55 | 69.23 |

Table 18: Results after the test set is expanded.

### A.8 FUTURE WORKS

This paper presents a preliminary exploration of the inductive reasoning capabilities of LLMs. We construct a post-training dataset for inductive reasoning based on number sequences, and after training, the LLMs show significant improvement in their inductive reasoning ability. Research on inductive reasoning in LLMs is still relatively limited in the academic community. Given the strong alignment between inductive reasoning and human learning paradigms, we believe this is a highly promising direction for future work.

**Inductive Reasoning Tasks, Benchmarks and Data**  This paper introduces the GTG task and the next number prediction task based on number sequences to evaluate the inductive reasoning ability of LLMs. However, using number sequences as the sole data source is relatively narrow. The inductive reasoning capability of LLMs can be examined from various other perspectives, such as format imitation (Cheng et al., 2024), cross-domain induction (Chen et al., 2024a), and multimodal inductive (Wang et al., 2024c), and so on. Our synthetic data is simple and easily extensible. Even a single data point has the potential to be expanded into either SFT or RL data. This serves as a concrete example demonstrating the scalability of inductive reasoning data. Based on our approach, we can generalize to a broader set of inductive reasoning tasks, benchmarks, and data, as long as a specific sandbox (or its alternative) can be constructed.

**Enhancing the Inductive Reasoning Ability of LLMs**  This paper enhances the inductive reasoning abilities of LLMs in the domain of number sequences through the use of synthetic data. It primarily

encourages LLMs to think by constructing cases, thereby improving their accuracy on specific tasks. However, it may be possible to design more sophisticated reward functions to explicitly guide LLMs to generate cases as a form of reasoning. Meanwhile, beyond case construction, we could incorporate additional process supervision signals (Cai et al., 2024a) into inductive reasoning tasks—for instance, more fine-grained steps such as "observation" (Sun et al., 2024) and "summarizing common patterns" (Siledar et al., 2024).

**Applications of Inductive Reasoning**     First, inductive reasoning is not limited to the text modality—it can also be applied to multimodal learning (Pan et al., 2025). Second, as a highly general and universal learning paradigm, inductive reasoning can play a significant role in AI education, enabling LLMs to guide teaching and communication (Dan et al., 2024; Xu et al., 2025) through analogy. Moreover, inductive reasoning can also fancilitate interdisciplinary collaboration—for example, knowledge from physics (Liu et al., 2025) can be used to solve problems in biology (Zhang et al., 2024a).

