# OpenReview forum: "Code-driven Number Sequence Calculation: Enhancing the Inductive Reasoning Abilities of Large Language Models"
_ICLR.cc/2026/Conference — Submitted to ICLR 2026_

### Official Review · Reviewer_FLvu · 2025-11-03

**Soundness:** 3
**Presentation:** 2
**Contribution:** 3
**Rating:** 4
**Confidence:** 2

**Summary:**

This paper addresses the deficiency of LLMs in inductive reasoning by introducing CodeSeq, a synthetic post-training dataset built from number sequences. The authors package number sequences into algorithmic problems to discover their general terms, defining a General Term Generation (GTG) task. Their pipeline generates supervised fine-tuning data through case-based reflection and iterative corrections, teaching LLMs autonomous case generation and self-checking. For reinforcement learning, they propose a Case-Synergy Solvability Scaling Reward (CSSR) that combines problem solvability (estimated from pass rate) and self-directed case generation success.

**Strengths:**

- First work to systematically use number sequences for LLM inductive reasoning training
- Comprehensive experiments on two base models (LLaMA3-8B, Qwen2.5-7B) with multiple random seeds
- Clear motivation for using number sequences and code-based representation
- Demonstrates substantial in-domain improvements

**Weaknesses:**

- No theoretical or empirical comparison showing why number sequences are fundamentally better for inductive reasoning
- Only 200 test samples for GTG evaluation - may not be statistically robust
- Data contamination risk: Number sequences from OEIS might overlap with pretraining data

**Questions:**

- What is the theoretical or empirical evidence that number sequences are superior to List Functions or ARC for inductive reasoning?
- How do you ensure number sequences from OEIS haven't been seen during pretraining?
- Why only 200 test samples for GTG? This seems insufficient for robust evaluation.
- The OOD improvements are very small. Are these statistically significant?

---

> ### Author Response · Authors · 2025-11-21
> **response to reviewer FLvu #1**
>
> We sincerely thank you for your valuable comments to help improve our paper.
>
> *For Weakness and Question "theoretical or empirical evidence that number sequences are superior"*:
> - We can demonstrate through experiments that number sequences provide greater benefits for inductive reasoning than List Functions or ARC data. We train LLaMA3-8B using both the data from [7] (data constructed from List Functions and ARC) and our CodeSeq. To ensure a fair comparison and prevent data leakage (since GTG, List, and ARC are included in the aforementioned training data), we evaluate the models on another inductive dataset [8] and the OOD task MMLU. The results in Table A and Table D below show that our data is **more effective on inductive tasks compared to previous training data** and **has transferability on other inductive benchmarks**, while maintaining better OOD performance.
>
> **Table A**
> |  | **[8]** |**MMLU**|
> |-----------------|--------|------|
> | **LLaMA3-8B** | 0.35| 0.62 |
> | **w/ [7]** | 0.35| 0.60 |
> | **w/ CodeSeq** | 0.37| 0.63 |
>
> - Compared with common visual or symbolic inductive tasks (such as ARC or list-function tasks), number-sequence tasks exhibit **deeper structural complexity** and a more abstract pattern-representation capability in inductive reasoning. Here is some evidence.
>   - First, the regularities in number sequences often involve multi-level recursions or nonlinear operations, resulting in a depth of pattern structure that far exceeds pixel-based transformations or simple logical mappings in ARC tasks [1].
>   - Second, the generative rules of number sequences can form complex inductive structures through combinations of operations, periodic nesting, and hierarchical dependencies, requiring models not only to capture explicit patterns but also to internalize the underlying generative mechanisms [2].
>   - LLMs perform well on short-term pattern recognition but show a significant drop in accuracy when dealing with complex recursions and closed-form inference, indicating that number-sequence tasks are more challenging at the abstract reasoning level [3].
>   - In contrast, although ARC and List Functions contain visual transformations, their rule space is discrete and low-dimensional, lacking the deeply structured regularities that can be continuously modeled in numerical sequences. Therefore, in terms of inductive structural depth, generative complexity, and abstraction of reasoning, number-sequence tasks represent a higher level of inductive difficulty.
> - Intuitively, ARC or List Function tasks focus on surface features such as shapes across cases, whereas number sequence tasks require delving into the underlying computational logic between items.
> - We also conduct experiments on GTG, ARC, and List Function tasks using two LLMs, evaluating all tasks with the same settings (that is, directly providing the number sequence itself and requiring the model to output the general term in code form). The results in Table B show that **number sequence tasks are relatively more difficult**, indicating that they inherently involve more complex underlying patterns.
>
> **Table B**
> |             | **GTG** | **ARC**  | **ListFunction** |
> |-----------------|--------|------|------|
> | **o3-mini-high**     | 0.28   | 0.35 | 0.55 |
> | **Claude-Sonnet-4**  | 0.21   | 0.24 | 0.52 |
>
> *For Weaknesses and Question "ensure number sequences from OEIS haven't been seen during pretraining"*:
> - **We answer this question by evaluating multiple strong LLMs on number-sequence tasks**.
>   - First, as shown in Table 2, even when sampling 32 answers, DeepSeek-R1-32B still scores below 0.3 on the GTG task, a result that far surpasses that of Qwen2.5-7B and LLaMA3-8B.
>   - Second, in Table 4, we evaluate the three strongest reasoning models on the next number prediction task (which can be seen as a simplified version of computing the general term), and the best result is still only around 0.4.
>   - Third, in the above Table B, the two strongest existing reasoning LLMs achieve less than 0.3 accuracy on the GTG task (here using the setup of directly inputting the number sequences themselves and sampling the answer once).
>
> *For Weaknesses and Question "only 200 test samples"*:
> - We thank the reviewer for the suggestion. We **expand the test set to 1,000 test** cases and show the updated results in Table C below, and the conclusions are consistent with those in the original Table 2: the CodeSeq training set indeed enhances LLMs’ inductive reasoning performance on new number sequences.
> - We use 200 test samples because some commonly used LLM benchmarks[4], and even some inductive reasoning benchmarks[5][6], have evaluation sample sizes ranging from 100 to 200, and our setup aligns with this scale.
> - Our training set contains approximately 1,000 number sequence patterns for both the SFT and RL portions, so using 200 number sequence patterns in our test set represents a reasonable proportion.

---

> ### Author Response · Authors · 2025-11-21
> **response to reviewer FLvu #2**
>
> **Table C**
> |             | **GTG(200)** | **GTG(1000)**  |
> |-----------------|--------|------|
> | **LLaMA3-8B**     | 9.27   | 8.30 |
> | **w/ CodeSeq**     | 44.22   | 44.89 |
> | **Qwen2.5-7B**  |  26.89  | 27.15  |
> | **w/ CodeSeq**  |  69.55  | 69.23 |
>
> *For Weaknesses and Question "OOD improvements are very small"*:
> - The results in Table D below (CodeSeq-trained Qwen on two other OOD inductive tasks) and Table A above indicate that our data **improves performance on general inductive reasoning tasks**, including ARC, ListFunctions, and the test dataset from [8], clearly demonstrating the usefulness of our data.
> - OOD benchmarks in Table 2 evaluate the LLMs’ general knowledge or capabilities, rather than other inductive reasoning benchmarks. As we describe in Section 4.3, our OOD benchmarks are intended to show that training the model with our data **does not compromise its general capabilities**, but rather necessarily improves them. This showcases the robustness of our data.
>
> **Table D**
> |  | **ARC** |**ListFunction**|
> |-----------------|--------|------|
> | **Qwen2.5-7B** | 0.00| 0.25 |
> | **w/ CodeSeq** | 0.01| 0.28 |
>
> **References**
>
> [1] Cognitive dimensions of numerical rule induction
>
> [2] Comparing Computer Models Solving Number Series Problems
>
> [3] Leveraging the Inductive Bias of Large Language Models for Abstract Textual Reasoning
>
> [4] Evaluating Large Language Models Trained on Code
>
> [5] A large-scale benchmark for few-shot program induction and synthesis
>
> [6] Neuro-Symbolic Hierarchical Rule Induction
>
> [7] MIRAGE: EVALUATING AND EXPLAINING INDUCTIVE REASONING PROCESS IN LANGUAGE MODELS
>
> [8] Language Models as Inductive Reasoners

---

### Official Review · Reviewer_aQXx · 2025-11-03

**Soundness:** 3
**Presentation:** 3
**Contribution:** 3
**Rating:** 6
**Confidence:** 2

**Summary:**

This paper addresses the challenge of enhancing inductive reasoning capabilities in large language models (LLMs) by introducing CodeSeq, a synthetic post-training dataset built from number sequences. The authors argue that existing inductive reasoning data focuses on superficial regularities and lacks complex internal patterns. To address this, they package number sequences into algorithmic problems aimed at discovering general terms, defining a General Term Generation (GTG) task. The pipeline generates supervised finetuning (SFT) data through case-based reflection on failed test cases with iterative corrections, teaching LLMs autonomous case generation and self-checking. Additionally, they employ reinforcement learning (RL) with a novel Case-Synergy Solvability Scaling Reward (CSSR) that considers both problem solvability (estimated from pass rates) and success rates of self-directed case generation. Experimental results demonstrate that models trained with CodeSeq improve on various reasoning tasks while preserving out-of-domain (OOD) performance.

**Strengths:**

- Novel approach of using number sequences as a source for inductive reasoning training data, which provides more complex internal patterns compared to existing datasets like List Functions or ARC
- Comprehensive experimental evaluation across 7 benchmarks (1 in-domain, 3 close-domain, 3 out-of-domain)
- Clear problem motivation highlighting two key challenges in inductive reasoning research

**Weaknesses:**

- The paper equates inductive reasoning primarily with number sequence pattern recognition, which is a narrow interpretation of inductive reasoning
- No discussion of potential data leakage or contamination from web-scraped number sequences
- No analysis of failure cases or error patterns to understand what the model still cannot solve

**Questions:**

- What percentage of problems required multiple reflection rounds, and what is the distribution?
- Have you tested on longer sequences or sequences with more complex patterns?

---

> ### Author Response · Authors · 2025-11-21
> **response to reviewer aQXx #1**
>
> We sincerely thank you for the effort and expertise that you contributed towards reviewing our paper.
>
> *For Weakness "number sequence is a narrow interpretation"*:
> - Compared to other inductive reasoning data, number sequences exhibit **excellent scalability and high verifiability**, and thus can be leveraged to produce synthetic data automatically. Other types of data are often **difficult to construct and require human labor**.
>   - Graph-structured data may not be easily convertible into textual forms familiar to LLMs [1], and graph rules cannot be executed and verified in the same straightforward way as code.
>   - Textual induction is susceptible to context, and multiple answers may be equally valid. Pure semantic matching is neither as rigorous nor as convenient as executable code for verification [2].
>   - Puzzle-style problems are feasible, but as puzzle complexity increases (e.g., in terms of the number of constraints or the scale of variables), the cost of expansion and verification rises significantly [3].
> - Compared with other inductive tasks, number-sequence tasks exhibit **deeper structural complexity** and a more abstract pattern-representation capability in inductive reasoning.
>   - First, the regularities in number sequences often involve multi-level recursions or nonlinear operations, resulting in a depth of pattern structure that far exceeds pixel-based transformations or simple logical mappings in ARC tasks [4].
>   - Second, the generative rules of number sequences can form complex inductive structures through combinations of operations, periodic nesting, and hierarchical dependencies, requiring models not only to capture explicit patterns but also to internalize the underlying generative mechanisms [5].
>   - LLMs perform well on short-term pattern recognition but show a significant drop in accuracy when dealing with complex recursions and closed-form inference, indicating that number-sequence tasks are more challenging at the abstract reasoning level [6].
> - In short, the essence of inductive reasoning is discovering patterns. Compared with other data that exhibit only a single form of regularity, number sequences feature nested and progressive patterns, making them more complex and closer to real-world scenarios (as detailed in Appendix A.3).
> - Our Codeseq not only performs well on the in-domain GTG task (Table 2) but also demonstrates **transferability on other inductive benchmarks** (Table A below), indicating that our data involves more complex underlying patterns.
>
> **Table A**
>
> |  | **ARC** |**ListFunction**|
> |-----------------|--------|------|
> | **Qwen2.5-7B** | 0.00| 0.25 |
> | **w/ CodeSeq** | 0.01| 0.28 |
>
> *For Weakness "data leakage"*:
> - We answer this question by **evaluating multiple strong LLMs on number-sequence tasks**.
>   - First, as shown in Table 2, even when sampling 32 answers, DeepSeek-R1-32B still scores below 0.3 on the GTG task, a result that far surpasses that of Qwen2.5-7B and LLaMA3-8B.
>   - Second, in Table 4, we evaluate the three strongest reasoning models on the next number prediction task (which can be seen as a simplified version of computing the general term), and the best result is still only around 0.4.
>   - Third, in Table B below, the two strongest existing reasoning LLMs achieve less than 0.3 accuracy on the GTG task (here using the setup of directly inputting the number sequences themselves and sampling the answer once).
>
> **Table B**
> |             | **GTG** |
> |-----------------|--------|
> | **o3-mini-high**     | 0.28   |
> | **Claude-Sonnet-4**  | 0.21   |
>
> *For Weakness "no analysis of failure cases or error patterns"*:
> - Thank you for the reviewer’s comments. We conduct a statistical analysis and investigation on the cases where the Qwen2.5 model fails both before and after training.
>   - We find that the majority of failures (about 60%) are due to the general term formulas of the number sequences involving **compositional rules**. For example, the sequence 10, 3, 2, 2, 3, 3, 3, …, is generated by a formula that combines square root operations, recursion, and modulo operations.
>   - Another portion (about 30%) of the number sequences contains **hidden variables**, where the sequence alone is insufficient to infer the rule, requiring reconstruction of the hidden state. For instance, the sequence 1, 1, 2, 4, 3, 7, 5, 12, 7, … actually represents a robot moving in the directions right, down, left, and up in order, with each step’s displacement equal to the sum of the previous two steps. The general term of the sequence corresponds to the robot’s Manhattan distance (the hidden variable in this case) from the starting point.
>   - These kinds of patterns make it challenging for LLMs to learn.

---

> ### Author Response · Authors · 2025-11-21
> **response to reviewer aQXx #2**
>
> *For Question "percentage and distribution of problems required multiple reflection rounds"*:
> - As mentioned above, number sequences that involve compositional rules and contain hidden variables are challenging to learn for LLMs, often requiring multiple reflection rounds to arrive at the correct solution.
> - In Table 1, we report the frequency of using reflection to obtain the final correct result. In the training data, the model performs 0 to 5 rounds of reflection, with an average of **2.56 rounds** of reflective reasoning needed to produce code that can pass all example tests. We present the **distribution of the number of revision rounds in Table 13**.
>
> *For Question "test on longer sequences or sequences with more complex patterns"*:
> - In the GTG task, each term of a number sequence may correspond to a test case. For each number sequence, we randomly select 5–7 terms as test cases.
> - Based on your suggestion, we select **50 number sequences of length greater than 10** and use all of their terms as test cases. The experimental results show that the code solutions for these 50 algorithmic problems pass all corresponding test cases. This confirms **the correctness of the code solutions produced by our pipeline**.
> - We thank the reviewer for the suggestion. We **expand the test set to 1,000 test cases to cover more complex patterns** and rerun the evaluation. The results are shown in Table C below, and the conclusions are consistent with those in the original Table 2: the CodeSeq training set indeed enhances LLMs’ inductive reasoning performance on new number sequences.
>
>  **Table C**
> |             | **GTG(200)** | **GTG(1000)**  |
> |-----------------|--------|------|
> | **LLaMA3-8B**     | 9.27   | 8.30 |
> | **w/ CodeSeq**     | 44.22   | 44.89 |
> | **Qwen2.5-7B**  |  26.89  | 27.15  |
> | **w/ CodeSeq**  |  69.55  | 69.23 |
>
> **References**
>
> [1] GRAPHLLM: BOOSTING GRAPH REASONING ABILITY OF LARGE LANGUAGE MODEL
>
> [2] CodeScore: Evaluating Code Generation by Learning Code Execution
>
> [3] ZebraLogic: On the Scaling Limits of LLMs for Logical Reasoning
>
> [4] Cognitive dimensions of numerical rule induction
>
> [5] Comparing Computer Models Solving Number Series Problems
>
> [6] Leveraging the Inductive Bias of Large Language Models for Abstract Textual Reasoning

---

### Official Review · Reviewer_kxad · 2025-11-04

**Soundness:** 2
**Presentation:** 2
**Contribution:** 2
**Rating:** 4
**Confidence:** 4

**Summary:**

This paper introduces a data pipeline called 'CodeSeq' that aims to enhance the reasoning abilities of LLMs by training them to infer general rules from number sequences. Therefore they transform number sequences into algorithmic problems expressed in code, enabling models to generate and test general terms through executable programs rather than abstract formulas. CodeSeq consists of three stages: (1) Sequence Algorithmization: converts curated number sequences into solvable algorithmic problems; (2) Case-based Reflection Injection: creates supervised finetuning data by iteratively refining code through test-based self-correction; and (3) Solvability-Estimated Selection: constructs reinforcement learning data using a new reward function prioritizes challenging problems and successful self-directed case generation. Experiments on open models such as  LLaMA3-8B and Qwen2.5-7B show that this appraoch improves reasoning and code reasoning performance without harming out-of-domain general abilities. Smaller models trained on CodeSeq achieve results comparable to much larger models.

**Strengths:**

Data pipeline
* The pipeline is conceptually thorough and the two-agent system (working and guiding LLMs) gives a built-in mechanism for problem validation, which improves data reliability.
* The sandbox testing ensures verification of code correctness.
* The CSSR reward is well designed and balances solvability and autonomous case-generation.

Experiments
* The experiments include ID, close-domain, and OOD tasks and hence give a comprehensive evaluation of reasoning improvements.
* Results show that CodeSeq improves inductive reasoning without sacrificing general reasoning.
* Multiple ablation variants help isolating which components matter, showing the importance of both reflection and solvability-based rewards.
* They further explore scaling effects (reflection rounds, pattern diversity & offer insights into data complexity and diminishing returns.
* The use of multiple seeds, multiple backbones, and standardized benchmarks (HumanEval, MMLU, etc.) adds credibility and reproducibility.

**Weaknesses:**

Data pipeline
* The section lacks quantitative examples.
* Can you please explain these manually written rules and give examples for them: "We first manually write rules to remove sequences with insufficient information, such as those without the calculation process or evolve from other sequences (requiring additional webpage links for references)."
* The reliance on LLM agents for data verification could introduce systematic errors > was this analysed? Where human evaluators invovled in verifying the success rate of the pipeline?
* The LLM-based reflection and validation process might be  expensive, pls discuss cost factors and tradeoffs between having a human-in-the-loop data gen approach vs purely synthetic with LLMs.
* There is no analysis of failure cases, e.g. algorithmic problems or code solutions being incorrect while passing validation.
* The paper provides limited evidence on the quality or representativeness of the final dataset.
* How reproducible is the pipeline? Given that it depends on specific prompt-engineering choices and models?

Experiments
* Eval focuses mostly on open-source models, specialized reasoning models (e.g. for Math or coding) would strengthen claims.
* Since number sequences are scraped from public sources, overlap between training and evaluation sequences might occur.
* While results show models improve, they don't qualitatively show with examples how model's reasoning chains improved before and after.

**Questions:**

See section weaknesses above

---

> ### Author Response · Authors · 2025-11-21
> **response to reviewer kxad #1**
>
> We sincerely appreciate your thorough and thoughtful comments, which are extremely valuable to our work.
>
> *For Weakness "lacks quantitative examples"*:
> - We provide an example of an algorithmic problem in Figure 10, and examples of the SFT data in Figures 16 and 17. For the RL data, we use the same input format as in Figure 16.
> - All training and test data are included in the **supplementary materials**.
>
> *For Weakness "manually written rules"*:
> - In Appendix A.4 (L1112–L1119), we explain in detail which crawled number sequences are removed by manual rules. Specifically:
>   - Some number sequence webpages do not provide any description or explanation of the current sequence and only give a link to the sequence’s original source. Since we do not perform nested crawling (i.e., following external links), we remove these webpages. **The lack of information or explanation prevents us from correctly converting the number sequences into algorithmic problems**.
>   - Number sequences with **too few terms** to form two example cases or 5–7 test cases are also removed.
>
> *For Weakness "data verification"*:
> - Our pipeline **minimizes bias as much as possible**.
>   - Length bias. The length of a sequence is independent of the data: each term in the sequence corresponds to one case in the algorithmic problem (Figure 1). We ensure that every problem contains exactly two example cases and five to seven test cases, and this quantity is strictly controlled.
>   - Pattern bias. Each number sequence corresponds to a distinct inductive pattern. We ensure that each pattern appears an equal number in the training or test sets, preventing the post-training model or the test distribution from being skewed toward any particular pattern.
> - Although our pipeline involves LLMs, we ensure data correctness through **human-written Python rules to eliminate any noise** (Appendix L1219–L1231).
>   - The High-Density Sequence Data already includes background and mathematical descriptions, so the working agent only needs to wrap the descriptions into a story.
>   - For generated example cases, human-written rules check if they exist in the original number sequence and compute the offsets.
>   - A guiding agent generates answers from the example cases' input and problem descriptions, keeping only correct ones to ensure consistency between the problem descriptions and the example cases.
>   - To avoid bias or limited diversity, different agents are used for problem generation and solving. We sample each problem and its answer multiple times to ensure diversity.
> - We run all code in a **sandbox unit test independent of the LLM**, ensuring that every case executes correctly. This guarantees that the intermediate processes are noise-free and that the final ground-truth answers are completely accurate.
> - Through the rigorous steps described above, we can ensure consistency between the process and the results, which in turn allows us to:
>   - Guarantee that each problem is correct.
>   - Achieve rich diversity in problem samples (more than 2,500 number sequence patterns with various sampling strategies).
>   - Avoid superficial edits: the correctness of the final answers shows that the LLMs engage in a natural and proper reflection process.
>   - Retain the LLM’s natural reflection process, training LLMs to generate cases automatically.
>
> *For Weakness "reflection and validation process might be expensive"*:
> - As mentioned in lines 1563–1567 of Appendix A.5, generating the data and testing via API (e.g., the results in Tables 3 and 4) cost a total of **$2,500 USD and ¥400 RMB**.
> - This entire pipeline can also be carried out with open-source small models (e.g., 7B or 8B). However, during the reflection process, **small models are often unable to provide correct revisions for code that executes incorrectly**, which reduces the efficiency of data generation.
> - Using a **human-in-the-loop setup** can also complete the same steps, but it requires **substantial human labor and time**, and it cannot replicate the naturally generated reflection process of an LLM.
> - In Table 1, we report the frequency of using reflection to obtain the final correct result. In the training data, the model performs 0 to 5 rounds of reflection, with **an average of 2.56 rounds** of reflective reasoning needed to produce code that can pass all example tests. We present the **distribution of the number of revision rounds in Table 13**.
> - To improve time efficiency, we conduct high-concurrency data construction. The total runtime of the entire pipeline code is approximately **50 hours**. Meanwhile, we also report the training efficiency of the data in Table 12.
> - During the reflection process, we strictly control the number of reflection rounds. Specifically, if the correct answer is not obtained after five rounds, we stop the process. This further improves the efficiency of our data generation.

---

> ### Author Response · Authors · 2025-11-21
> **response to reviewer kxad #2**
>
> *For Weakness "algorithmic problems or code solutions being incorrect while passing validation"*:
> - We ensure the correctness of the algorithmic problems themselves through **a rigorous procedure** (Appendix L1219–L1231).
>   - The High-Density Sequence Data already includes background and mathematical descriptions, so the working agent only needs to wrap the logic into a story.
>   - For generated examples, human-written rules check if they exist in the original sequence and compute the offsets.
>   - A guiding agent generates answers from the examples and problem description, keeping only correct ones to ensure consistency between the problem descriptions and the example cases.
>   - To avoid bias or limited diversity, different agents are used for problem generation and solving. We sample each problem and its answer multiple times to ensure diversity.
> - If a code solution passes the test case verification, it is guaranteed to be correct.
>   - The code solution is executed and validated on cases through **unit tests** independent of the LLM.
>   - As stated in lines 1283–1285 of Appendix A.4.3, all settings for our code unit tests are derived from the general evaluation framework OpenCompass. The code unit tests within this framework are designed to **assess code tasks across various scenarios, and are therefore inherently intended for general tasks**. We use exactly the same settings, so there is no risk that the tests provide shortcuts.
>   - We select 50 number sequences of length greater than 10 and use all of their terms as test cases. The experimental results show that **the code solutions for these 50 algorithmic problems pass all corresponding test cases**. This confirms the correctness of the code solutions produced by our pipeline.
>
> *For Weakness "evidence on the quality or representativeness of the dataset"*:
> - Through various experiments, we demonstrated the relatively strong performance of our data across several tasks: the in-domain GTG task, other inductive reasoning tasks, three closed-domain code tasks, and the OOD reasoning task. Therefore, the data possesses a certain degree of representativeness.
>   - The GTG column in Table 2 shows that our synthetic data performs well on in-domain tasks.
>   - On the three code-related tasks in Table 2, our data also exhibits strong transferability.
>   - On three OOD benchmarks, our data can maintain its performance, demonstrating robustness.
>   - Table A below presents the results of our CodeSeq-trained Qwen2.5-7B on two other types of inductive reasoning tasks. These results reflect that our data effectively **enhances the model’s inherent inductive capabilities**.
>   - We compare our data with another simple inductive training dataset [1] in Table B below, demonstrating that our training data incorporates human case-driven reasoning and can be used for training via both SFT and RL.
>   - We train LLaMA3-8B using both the data from [1] and our CodeSeq. To ensure a fair comparison and prevent data leakage (since GTG, List, and ARC are included in the aforementioned training data), we evaluate the models on another inductive dataset [2] and the OOD task MMLU. The results in Table C show that our data is **more effective on inductive tasks compared to previous training data, while maintaining better OOD performance**.
>   - We validate the robustness of our data across different models, including Qwen2.5-7B (Table 2), LLaMA3-8B (Table 2), Qwen2.5-32B (Table 9), and two well-known open-source LLMs (with 8B and 200B+ versions). We also compare different RL algorithms, such as PPO, DPO, and DAPO (Table 11), as well as various reward functions (Appendix A.7.4). These results collectively demonstrate the effectiveness of our algorithm and data.
>
>   **Table A**
>
> |  | **ARC** |**ListFunction**|
> |-----------------|--------|------|
> | **Qwen2.5-7B** | 0.00| 0.25 |
> | **w/ CodeSeq** | 0.01| 0.28 |
>
>   **Table B**
> |  | **[1]**                 | **CodeSeq**             |
> |------------------|--------------------------|--------------------------|
> | **Data Construction** | rule-based construction | synthetic data          |
> | **Training Method**   | SFT                      | SFT and RL              |
> | **Prompt Formation**  | ICL                      | case-driven reflection  |
> | **Difficulty Control**| No                       | Yes                     |
>
>   **Table C**
> |  | **[2]** |**MMLU**|
> |-----------------|--------|------|
> | **LLaMA3-8B** | 0.35| 0.62 |
> | **w/ [1]** | 0.35| 0.60 |
> | **w/ CodeSeq** | 0.37| 0.63 |

---

> ### Author Response · Authors · 2025-11-21
> **response to reviewer kxad #3**
>
> - We can explain the quality of our data by looking at its accuracy, diversity, and the characteristics of the number sequences.
>   - In the section titled 'algorithmic problems or code solutions being incorrect while passing validation' above, we strictly detail how our data ensures the **correctness of both the algorithmic problems and the code solutions**.
>   - There is **no overlap between the training and test sets**. Each number sequence represents a distinct inductive pattern, and to cover as many inductive patterns as possible, every data point in our SFT training set, RL training set, and GTG test set is unique with no repetition. As a result, the model encounters entirely new sequences during the testing phase. This demonstrates the diversity of our data.
>   - Compared with common visual or symbolic inductive tasks (such as ARC or list-function tasks), number-sequence tasks exhibit **deeper structural complexity** and a more abstract pattern-representation capability in inductive reasoning. Here is some evidence.
>     - First, the regularities in number sequences often involve multi-level recursions or nonlinear operations, resulting in a depth of pattern structure that far exceeds pixel-based transformations or simple logical mappings in ARC tasks [3].
>     - Second, the generative rules of number sequences can form complex inductive structures through combinations of operations, periodic nesting, and hierarchical dependencies, requiring models not only to capture explicit patterns but also to internalize the underlying generative mechanisms [4].
>     - LLMs perform well on short-term pattern recognition but show a significant drop in accuracy when dealing with complex recursions and closed-form inference, indicating that number-sequence tasks are more challenging at the abstract reasoning level [5].
>     - Intuitively, ARC or List Function tasks focus on surface features such as shapes across cases, whereas number sequence tasks require delving into the underlying computational logic between items.
>
> *For Weakness " how reproducible is the pipeline"*:
> - We have already made the code for our data pipeline publicly available in an **anonymous repository** (see Abstract), and we have also provided the data and code in **supplementary material**.
> - The prompts we design are general and can be utilized by all models. However, we use more powerful LLMs to increase the **efficiency of data synthesis**. If smaller models are applied, achieving the same results might require more time and cost.
>
> *For Weakness "specialized reasoning models would strengthen claims"*:
> - Thank you very much for your suggestions. We conduct experiments and tests using the **Qwen2.5-Coder-7B** model. The results, presented in Table D below, show that our data shows the same outcomes as in Table 2 for reasoning models—that is, improvements on both the GTG and code tasks, while also preserving OOD performance.
>
> **Table D**
> | **qwen2.5-coder-7B**| **GTG**| **humaneval**|**mbpp**|**lcb**| **mmlu**| **bbh**| **gaokao**|
> | :---:        | :---: | :---:     | :---: | :---: | :---: | :---: | :---:  |
> | vanilla     | 30.11 | 88.41     | 78.99 | 44.57 | 69.91 | 67.38 | 67.54  |
> | w/ CodeSeq  | 70.05 | 89.93     | 79.55 | 45.60 | 69.82 | 67.11 | 67.58  |
>
> *For Weakness "overlap between training and evaluation"*:
> - There is no overlap between the training and test sets. Each sequence represents a distinct inductive pattern and corresponds to a unique ID; **simply deduplicating by IDs is sufficient to prevent any overlap**.  Every data point in our SFT training set, RL training set, and GTG test set is unique with no repetition. As a result, the model encounters entirely new sequences during the testing phase.
>
> *For Weakness "how model's reasoning chains improved before and after"*:
> - The experimental results in Figure 5 demonstrate that, after training, the model can **not only improve the correctness of constructing cases in CoT but also increase the final success rate in solving problems**. This part of the experiment demonstrates that the model can, to some extent, learn to self-generate cases, thereby improving the hit rate of its generated hypotheses.
> -  The model before training might generate fake cases and produce incorrect code solutions. After training, the model can generate realistic analogous cases, thereby improving the correctness of the code solutions. For example, for the number sequence 2, 5, 3, 10, 2, 5, 3, 10, …, its general term involves square recursion and modulo operations. For the base model, the case it generates could be (4, 9)—that is, the model predicts the fourth term as 9, whereas the correct fourth term is 10. After training, this problem no longer occurs.

---

> ### Author Response · Authors · 2025-11-21
> **response to reviewer kxad #4**
>
> **References**
>
> [1] MIRAGE: EVALUATING AND EXPLAINING INDUCTIVE REASONING PROCESS IN LANGUAGE MODELS
>
> [2] Language Models as Inductive Reasoners
>
> [3] Cognitive dimensions of numerical rule induction
>
> [4] Comparing Computer Models Solving Number Series Problems
>
> [5] Leveraging the Inductive Bias of Large Language Models for Abstract Textual Reasoning

---

### Official Review · Reviewer_iUqk · 2025-11-04

**Soundness:** 3
**Presentation:** 3
**Contribution:** 3
**Rating:** 6
**Confidence:** 3

**Summary:**

This work focuses on improving inductive reasoning capability in LLMs. The authors first package number sequences into algorithmic problems to discover their general terms, defining a general term generation (GTG) task that can be represented by code solutions. Then, the authors propose a synthetic data pipeline, forming a post-training dataset that consists of SFT and RL data for solving number sequence algorithmic problems.

**Strengths:**

1. This work enhances inductive reasoning ability in LLMs, which is an unexplored yet interesting direction. The General Term Generation (GTG) task is well-motivated and addresses a critical gap in inductive reasoning benchmarks.

2. Packaging number sequences as algorithmic problems with code solutions is novel, and the number sequence data synthetic pipeline is methodologically sound.

3. The synthetic data CodeSeq is proven effective for various reasoning tasks and achieves good performance on the proposed GTG task.

**Weaknesses:**

### Main weakness

- One concern is the scope between sequence data and inductive reasoning definition: Framing inductive reasoning as general term generation in number sequences is reasonable as a starting point but risks oversimplification. It is debatable to what extent sequence prediction captures "true" inductive reasoning as done in broader sentries (*e.g.*, relational, visual, or linguistic domains).

- Limited evaluation benchmarks. The improvements in inductive reasoning ability are not benchmarked on existing public datasets but only evaluated on a single self-crafted GTG subset. I strongly suggest the authors provide results on CodeARC benchmark [1] (Wei et al., 2025) to further validate the effectiveness of the proposal method.

- Unclear costs. The data construction pipeline involves iterative generation and reflection using APIs of superior LLMs. I wonder if it's costly to generate such a dataset. I suggest the authors can clearly report relevant costs and time efficiency.


### Minor weakness
- Several relevant recent works and discussions [1][2] on inductive reasoning in LLMs are missing from the related work.

- Although the authors claim "no security concerns" in the ethics statement, it would be helpful to discuss about known risks of LLMs in the context of inductive reasoning, such as bias, deception, or producing wrong but convincing answers.

---

[1] CodeARC: Benchmarking Reasoning Capabilities of LLM Agents for Inductive Program Synthesis. arXiv preprint:2503.23145

[2] Patterns Over Principles: The Fragility of Inductive Reasoning in LLMs under Noisy Observations. arXiv preprint:2502.16169

**Questions:**

1. What guarantees does the use of code unit tests offer against "shortcut" code solutions that pass curated test cases but fail in other general cases? Has any formal error analysis been performed?

2. In reward function CSSR, the authors combine the metric of $ \frac{N_{tc}}{N_c} $. This seems to require the model to have the ability to generate test data in CoT itself. But after checking the $CodeSeq_{RL}$ training data, I don't see prompts that guide the model to do this. Could the author explain this formula?

---

> ### Author Response · Authors · 2025-11-21
> **response to reviewer iUqk #1**
>
> We are grateful for your careful review, helpful comments and detailed questions that ensued.
>
> *For Weakness "the scope between sequence data and inductive reasoning definition"*:
> - Our Codeseq not only performs well on the in-domain GTG task (Table 2) but also **demonstrates transferability in broader inductive tasks** (e.g., ListFunction for relational domain, ARC for visual domain). Results in Table A below indicate that our data enhances basic inductive reasoning capabilities to some extent, and therefore leads to improvements in other inductive domains.
>
> **Table A**
> |  | **ARC** |**ListFunction**|
> |-----------------|--------|------|
> | **Qwen2.5-7B** | 0.00| 0.25 |
> | **w/ CodeSeq** | 0.01| 0.28 |
>
> - Compared to other inductive reasoning data, number sequences exhibit excellent scalability and high verifiability, and thus can be leveraged to produce synthetic data automatically. **Other types of data are often difficult to construct and require human labor**.
>   - Most relational datasets rely on a limited and fixed set of operations (e.g., map/reduce/sort). Because the underlying rules are highly enumerable, models tend to memorize template-like mappings rather than truly learning inductive reasoning [1].
>   - Linguistic induction is highly sensitive to context, and multiple answers may be equally valid. Pure semantic matching is neither as rigorous nor as convenient as executable code for verification [2].
>   - Visual inductive problems are feasible, but as the complexity increases (e.g., in terms of the number of constraints or the scale of variables), the cost of expansion and verification rises significantly [3].
> - Compared with other inductive tasks, number-sequence tasks exhibit **deeper structural complexity** and a more abstract pattern-representation capability in inductive reasoning.
>   - First, the regularities in number sequences often involve multi-level recursions or nonlinear operations, resulting in a depth of pattern structure that far exceeds pixel-based transformations or simple logical mappings in ARC tasks [4].
>   - Second, the generative rules of number sequences can form complex inductive structures through combinations of operations, periodic nesting, and hierarchical dependencies, requiring models not only to capture explicit patterns but also to internalize the underlying generative mechanisms [5].
>   - LLMs perform well on short-term pattern recognition but show a significant drop in accuracy when dealing with complex recursions and closed-form inference, indicating that number-sequence tasks are more challenging at the abstract reasoning level [6].
> - In short, the essence of inductive reasoning is discovering patterns. Compared with other data that exhibit only a single form of regularity, number sequences feature nested and progressive patterns, making them more complex and closer to real-world scenarios (as detailed in Appendix A.3).
>
> *For Weakness "Limited evaluation benchmarks"*:
> - We evaluate **two other domains of inductive reasoning tasks** in Table A above.
> - Thank you very much for the suggestion. We evaluate the success rate on the Annotated Dataset of the **CodeARC benchmark** in Table B, and the results also show improvement, further demonstrating that our data enhances the inductive capabilities of LLMs.
>
> **Table B**
> |  | **CodeARC** |
> |-----------------|-----------------|
> | **Qwen2.5-7B** | 29.2|
> | **w/ CodeSeq** | 30.1|
>
> *For Weakness "Unclear costs"*:
> - As mentioned in lines 1563–1567 of Appendix A.5, generating the data and testing via API (e.g., the results in Tables 3 and 4) cost a total of **$2,500 USD and ¥400 RMB**.
> - To improve time efficiency, we conduct high-concurrency data construction. The total runtime of the entire pipeline code is approximately **50 hours**. Meanwhile, we also report the training efficiency of the data in Table 12.
> - During the reflection process, we strictly control the number of reflection rounds. Specifically, if the correct answer is not obtained after 5 rounds, we stop the process. The **average number of rounds is 2.56**. This further improves the efficiency of our data generation.
>
> *For Weakness "several relevant recent works are missing"*:
> - We sincerely thank the reviewer for the suggestion and will include these two papers in the revision.
>   - CodeARC is an inductive dataset derived from public code benchmarks, involving only simple SFT algorithms. In contrast, we use easily extendable, executable, and verifiable number sequence data that can support both SFT and RL training.
>   - Patterns Over Principles studies the robustness of inductive rules under noisy conditions, which focuses on a different aspect than our work.

---

> ### Author Response · Authors · 2025-11-21
> **response to reviewer iUqk #2**
>
> *For Weakness "discuss about known risks"*:
> - We sincerely thank the reviewer for the suggestion. Indeed, inductive reasoning carries certain risks.
>   - Confidently wrong outputs: When performing induction, LLMs may "hypothesize a pattern", generating a self-consistent but incorrect answer even when the data does not follow any true rule.
>   - Bias: If the training data itself is biased (e.g., some patterns appear more frequently), the model may tend to produce these common patterns, resulting in systematic bias.
> - However, in our work, we have taken steps to **mitigate such risks as much as possible**.
>   - First, the ground truth for all training data is verified using unit tests independent of any LLM, ensuring both answer correctness and process consistency.
>   - Each sample in our training and test sets is a unique number sequence, and we ensure that no number sequence is repeated. Moreover, each number sequence embodies a completely different inductive pattern, which helps to alleviate this type of bias.
> - We will list these potential risks in the next version, along with the steps we have taken to mitigate them.
>
> *For Question "shortcut code solutions"*:
> - As stated in lines 1283–1285 of Appendix A.4.3, all settings for our code unit tests are derived from the general evaluation framework OpenCompass. The code unit tests within this framework are designed to **assess code tasks across various scenarios, and are therefore inherently intended for general tasks**. We use exactly the same settings, so there is hardly any risk that the tests provide shortcuts specific to number-sequence tasks while failing on general tasks.
>
> *For Question "don't see prompts for RL"*:
> - In CodeSeqRL, we provide the algorithmic problems themselves. For the convenience of quickly switching data formats ( for example, to conduct ablation studies or analyze using different pass rate), the "generate test data" portion of the CSSR formula prompt is added in the post-processing script of the VERL training framework.
> - We sincerely thank the reviewer for the suggestion and will include this prompt in the RL dataset in the revision.
>
> **References**
>
> [1] MIRAGE: EVALUATING AND EXPLAINING INDUCTIVE REASONING PROCESS IN LANGUAGE MODELS
>
> [2] CodeScore: Evaluating Code Generation by Learning Code Execution
>
> [3] ZebraLogic: On the Scaling Limits of LLMs for Logical Reasoning
>
> [4] Cognitive dimensions of numerical rule induction
>
> [5] Comparing Computer Models Solving Number Series Problems
>
> [6] Leveraging the Inductive Bias of Large Language Models for Abstract Textual Reasoning

---

### Official Review · Reviewer_Evb6 · 2025-11-04

**Soundness:** 3
**Presentation:** 2
**Contribution:** 1
**Rating:** 2
**Confidence:** 4

**Summary:**

The paper presents a framework and dataset to finetune models on inductive reasoning tasks, where the training data is based on number sequences. Number sequences are transformed into algorithmic problems, and the models are trained to produce code that provides a general term for the number sequence. A dataset for SFT and RL-based finetuning is constructed in several steps, letting a larger LM reason about the problem and make several attempts. Experiments show that finetuning with this dataset can improve the performance of small models significantly, sometimes even outperforming closed-source models.

**Strengths:**

1. The paper presents an interesting approach to convert simple number sequences into a large synthetic dataset with a variety of algorithmic problems.
2. There are extensive ablation studies that justify the design choices and chosen parameters.
3. The experiments show large performance improvements on related benchmarks.

**Weaknesses:**

1. The methodological novelty of the framework is limited, since it mainly consists of distilling knowledge from LLMs into smaller models, by letting the larger models construct algorithmic problems and solve them. Standard finetuning methods are used.
2. In general, the contribution is unclear. In the abstract, the contribution is framed as a dataset, but usually a dataset is proposed as a benchmark or for a specific real-world application, and not as a toy problem for finetuning. There are some nice contributions, such as defining a solvability score for problems and weighting the RL reward by the solvability score. If the main contribution are these methodological adaptations, it would be required to train on other datasets to show whether they consistently improve performance. If the main contribution is a dataset of toy problems, it would be necessary to show that this dataset is more useful for finetuning (or complementary) to other datasets.
3. The performance on out-of-domain problems only shows very limited improvements. However, if the models do not achieve good performance on general inductive reasoning tasks, it is questionable whether this method is useful.
4. The need for this dataset is motivated by saying that other related datasets such as ARC “fail to form complex internal patterns”. However, Table 3 shows that ARC is a lot harder for LLMs than the proposed dataset.

**Questions:**

1. I’m very confused about the notion of “publishing a RL dataset”. To the best of my knowledge, GRPO (and other common RL algorithms) are trained on-policy, so that the data is only generated during training. The reward is computed for the generated answer and then the model is updated. So how is it possible to construct CodeSeqRL dataset and train multiple models with it?
2. Why not compare to a model fine-tuned with another dataset, e.g. the train split of the Humaneval or MBPP dataset? This would show whether the proposed dataset brings an advantage compared to existing datasets.

---

> ### Author Response · Authors · 2025-11-21
> **response to reviewer Evb6 #1**
>
> We sincerely appreciate your detailed and specific feedback on the paper. Your suggestions will greatly enhance our paper. I hope to clarify and engage in further discussion with you.
>
> *For Weakness "standard finetuning methods are used"*:
> - The work in this paper **does not directly distill powerful LLMs**; rather, it merely has LLMs generate algorithmic problems based on number sequences and produce reflection content. The construction and validation of the synthetic data are carried out using human-designed rules and independent sandbox unit tests. The ablation experiments in Table 2 demonstrate that **the data, the CoT design, and the algorithms themselves are the key factors for success**.
> - Another reason we do not directly distill larger LLMs is that **their performance on number sequence problems is inherently limited**. Table 4 shows the results of the three most powerful reasoning models on the next number prediction task—a simplified version of the sequence general term task—which reflects that even the strongest models can only achieve around 0.4 accuracy.
> - Using powerful, closed-source LLMs to synthesize data is a standard practice in mainstream LLM training, such as Qwen3 [1] and Kimi K2 [2]. This is not one of our innovations.
> - Our main innovations are in the following aspects:
>   - We use number sequences, possessing deep implicit inductive patterns, as a source of inductive data. The practical role and advantages of number sequences are detailed in Appendix A.3.
>   - Building on this, we transform them into scalable and verifiable code-based forms, providing precise case-driven reflection and controllable difficulty, which enables LLMs to learn how to construct cases autonomously and enhance their inductive reasoning ability. To our knowledge, we are the first in the field of inductive reasoning to construct synthetic data (both SFT and RL) that makes models generate cases and perform reflection spontaneously.
> - Our data has already been applied to a well-known open-source LLM (with 8B and 200B+ versions).
>
> *For Weakness "the contribution is unclear, whether data or methodology"*:
> - Our main contribution lies in **proposing synthetic data for inductive reasoning**. It uses number sequences as the data source, injecting precise reflection mechanisms into the model, and implements difficulty control.
>   - In addition to the synthetic data for LLM training, we also provide a GTG test set in the same format and evaluate it across numerous models (Tables 2 and 3, etc.), which can likewise be regarded as a benchmark.
>   - Synthetic data and benchmarks based on number sequences have broad applications in real life (Appendix A.3).
>   - Moreover, compared with other types of inductive data, the patterns within number sequences often involve multi-level recursion or nonlinear operations. Their generative rules can form complex inductive structures through combinations of operations, periodic nesting, and hierarchical dependencies, making them a deeper and more sophisticated source of data with high information density.
> - Our method is built around the data, including the case-based reflection paradigm used in SFT and the CSSR reward function used in RL.
> - We demonstrate **the effectiveness of synthetic data through multiple experiments**.
>   - The GTG column in Table 2 shows that our synthetic data performs well on in-domain tasks.
>   - On the three code-related tasks in Table 2, our data also exhibits strong transferability.
>   - On three OOD benchmarks, our data can maintain models' performance, demonstrating robustness.
>   - Table A below presents the results of our CodeSeq-trained Qwen2.5-7B on **two other types of inductive reasoning tasks**. These results reflect that **our data effectively enhances the model’s inherent inductive capability**.
>   - We compare our data with another simple inductive training dataset [3] in Table B below, demonstrating that our training data incorporates human case-driven reasoning and can be used for training via both SFT and RL. It reflects the novelty of our data.
>   - We **train LLaMA3-8B using both the data from [3] and our CodeSeq**. To ensure a fair comparison and prevent data leakage (since GTG, List, and ARC are included in the aforementioned training data), we evaluate the model on another inductive dataset [4] and the OOD task MMLU. The results in Table C show that our data is more effective on inductive tasks, while maintaining better OOD performance.
>   - We validate the robustness of our data across different models, including Qwen2.5-7B (Table 2), LLaMA3-8B (Table 2), Qwen2.5-32B (Table 9), and two well-known open-source LLMs (with 8B and 200B+ versions). We also compare different RL algorithms, such as PPO, DPO, and DAPO (Table 11), as well as various reward functions (Appendix A.7.4). These results collectively demonstrate the effectiveness of our algorithm and data.

---

> ### Author Response · Authors · 2025-11-21
> **response to reviewer Evb6 #2**
>
> **Table A**
> |  | **ARC** |**ListFunction**|
> |-----------------|--------|------|
> | **Qwen2.5-7B** | 0.00| 0.25 |
> | **w/ CodeSeq** | 0.01| 0.28 |
>
> **Table B**
> |  | **[3]**                 | **CodeSeq**             |
> |------------------|--------------------------|--------------------------|
> | **Data Construction** | rule-based construction | synthetic data          |
> | **Training Method**   | SFT                      | SFT and RL              |
> | **Prompt Formation**  | ICL                      | case-driven reflection  |
> | **Difficulty Control**| No                       | Yes                     |
>
>
> **Table C**
> |  | **[4]** |**MMLU**|
> |-----------------|--------|------|
> | **LLaMA3-8B** | 0.35| 0.62 |
> | **w/ [3]** | 0.35| 0.60 |
> | **w/ CodeSeq** | 0.37| 0.63 |
>
> *For Weakness "performance on out-of-domain problems"*:
> - The results in Table A and Table C above indicate that our data **improves performance on general inductive reasoning tasks**, including ARC, ListFunctions, and the dataset from [4], clearly demonstrating the usefulness of our data.
> - OOD benchmarks in Table 2 evaluate the LLMs’ general knowledge or capabilities, rather than other inductive reasoning benchmarks. As we describe in Section 4.3, our OOD benchmarks are intended solely to show that training the model with our data **does not compromise its general capabilities**, rather than to necessarily improve them. This showcases the robustness of our data.
>
> *For Weakness "ARC is a lot harder"*:
> - In Table 3, the lower ARC scores are due to the use of **different metrics** compared to our GTG task. In our GTG task, we sample each test example 32 times with a high temperature, whereas the ARC task uses a low temperature and only a single sample (aligned with the official setting). This is one reason for the discrepancy.
> - In the GTG task, we wrap the number sequences into algorithmic problems. The problem descriptions contain **partial clues about the general terms**, which can increase the LLMs’ accuracy in computing the general terms.
> - Under **the same evaluation setting as ARC** (that is, directly providing the number sequence itself and requiring the model to output the general term in code form), the performance of two models on the GTG task is in Table D below. The results prove that GTG is a more challenging task.
> - In the prior experiments in Table 4 (Appendix A.2), we present the results of the next number prediction task, a simplified version of the GTG task. The results show that even powerful closed-source LLMs cannot achieve high scores on this task.
> - Overall, the GTG task is more difficult than the ARC task, precisely because it involves more complex internal inductive patterns.
>
> **Table D**
> |             | **GTG** | **Arc**  | **List** |
> |-----------------|--------|------|------|
> | **o3-mini-high**     | 0.28   | 0.35 | 0.55 |
> | **Claude-Sonnet-4**  | 0.21   | 0.24 | 0.52 |
>
> *For Problem "about GRPO"*:
> - By saying we "publish an RL dataset", we mean that **each problem in the RL training set is synthesized and filtered according to our pipeline**.
> - All your descriptions regarding GRPO are correct, and we strictly follow the GRPO algorithm during training.
> - CodeseqRL is only used as the training set for RL. Each sample includes a problem and several cases for testing the answers. During RL, only one model is trained (Qwen2.5-7B or LLaMA3-8B in Table 2). Each problem is rolled out multiple times during training, and rewards are computed accordingly. This process **strictly follows the GRPO algorithm**.
>
> *For Problem "compare with train split of the Humaneval or MBPP dataset"*:
> - MBPP and HumanEval are standard evaluation benchmarks. To ensure fairness and align with the LLM evaluation protocols used in industry, they cannot be split. Moreover, HumanEval contains only around 150 samples, so further splitting would not yield good performance.
> - However, thanks to the reviewer’s suggestion, we **use all the test data from HumanEval to train Qwen2.5-7B** and further continue training it on CodeSeq. The experimental results in Table E demonstrate that CodeSeq provides additional gains on inductive tasks compared to data like HumanEval.
> - We also **use all the test data from HumanEval for training and evaluating on MBPP and LCBench** (since these are all code-related tasks). The experimental results are shown in Table F below, which exhibits that our data is relatively challenging and of high quality for the base model in the code domain.
>
> **Table E**
> |  | **GTG** | **ARC** |
> |-----------------|--------|--------|
> | **Qwen2.5-7B** | 26.89 | 0.0 |
> | **w/ Heval**   | 26.93 | 0.0 |
> | **w/ Heval&CodeSeq**   | 69.51| 0.9|
>
> **Table F**
> |  | **MBPP** | **LCBench** |
> |-----------------|--------|--------|
> | **Qwen2.5-7B** | 71.59 | 41.97 |
> | **w/ Heval**   | 71.63| 42.21 |
> | **w/ CodeSeq**   | 75.49| 43.89|

---

> ### Author Response · Authors · 2025-11-21
> **response to reviewer Evb6 #3**
>
> **References**
>
> [1] Qwen3 Technical Report
>
> [2] Kimi K2: Open Agentic Intelligence
>
> [3] MIRAGE: EVALUATING AND EXPLAINING INDUCTIVE REASONING PROCESS IN LANGUAGE MODELS
>
> [4] Language Models as Inductive Reasoners

---

> > ### Comment · Reviewer_Evb6 · 2025-11-27
> >
> > Thank you for the thorough response. My questions regarding GRPO and contribution were clarified, and some new experiments were added showing advantages of the proposed synthetic data. Nevertheless, I still believe that the contribution is quite limited.
> > I understand that the "OOD benchmarks" might be too far from the original dataset, but if the new dataset only helps for improving the performance on toy data, it's basically useless. So unless there is some significant improvement on many related benchmarks, I don't see the need for this dataset. The problem is that most of the additional results show only very minor improvements on benchmarks. For example, ARC can be seen as the most related inductive reasoning benchmark; however, training on the proposed dataset only improves the performance from 0.0 to 0.01. If ARC is easier than the proposed data, and if the main contribution is a synthetic dataset for improving performance in inductive reasoning, why is the performance on ARC so bad? On most other benchmarks, the performance increases are very minor as well.
> >
> > Also, I'm surprised that in the new experiment where the proposed dataset is compared to finetuning with another dataset [3] (Table C), the other dataset basically has no effect, or even decreases performance. This does not seem very realistic.
> >
> > I will increase my score since many experiments were added which improve the paper, but I remain sceptical whether a toy dataset with small improvements justifies acceptance.

---

> ### Author Response · Authors · 2025-11-27
> **More Clarifications for Reviewer Evb6**
>
> We sincerely appreciate your willingness to increase the rating score since "many experiments were added which improve the paper". Below, we give more clarifications on your other concerns.
>
> *For "the new dataset only helps for improving the performance on toy data"*:
> - To further demonstrate the effectiveness of our data on some related data, we conduct tests on a **symbolic inductive reasoning task** [4] and a **program inductive task** [5] in the OOD setting with two models, as shown in Table G below.
> - The results in Table 2, Table E, and Table F demonstrate that our data exhibits strong transferability not only on toy tasks but also **on code generation tasks**, even some completely unrelated OOD knowledge tasks.
>
> **Table G**
> |  |**[4]**|**[5]**|
> |-----------------|--------|--------|
> | **Qwen2.5-7B** | 39.0 | 29.2 |
> | **w/ CodeSeq** |42.0 | 30.1 |
> | **LLaMA3-8B** | 35.0 | 16.1 |
> | **w/ CodeSeq** |37.0 |19.7 |
>
> *For "the improvement of ARC and other benchmarks is small"*:
> - As we mentioned in the "ARC is a lot harder" section, our GTG task uses **different testing conditions and metrics** compared to ARC. Under the same settings, the results in Table D demonstrate that GTG is still more challenging than ARC.
> - The results in Table 2, Table E, and Table F demonstrate that our data exhibits **strong transferability on code generation tasks**. Table G shows that our data brings an average improvement of about **2.5 points on code induction and symbolic induction tasks** in the OOD setting. Also, Qwen2.5-7B achieves an average improvement of about 2 points (approximately an **8% relative gain**) across four OOD benchmarks, which is significant for this scenarios.
> - Although both the ARC task and our GTG task fall under inductive reasoning, they differ significantly in form. The ARC task emphasizes the model’s ability to recognize visual or spatial similarities, whereas our task focuses on exploring the evolutionary logic between numbers.
> - After training on our data, the Qwen2.5-7B model is able to achieve performance comparable to models such as GPT-4.5 and Claude 3.7 on this task.
>
> *For "finetuning with another dataset [3] has no effect or even decreases performance"*:
> - Because the data in [3] are constructed from common puzzles, lists, strings, etc. These datasets, as typical LLM reasoning sources, are likely to **suffer from data leakage**—that is, the model may have already seen similar content during training, preventing further improvement in capability. Therefore, [3] cannot enhance OOD inductive reasoning ability (as shown by the results when testing with [4]).
> - For example, **the Intern series [9], as well as the Minimax-M1 [10] and Seed-thinking models [11]**, all make extensive use of large amounts of toy-style synthetic data, such as puzzles.
> - Moreover, numerous studies [6][7][8] have shown that **training on previously seen samples can cause overfitting and harm a model’s OOD capabilities** (as evidenced by the MMLU results).
> - The above discussion further illustrates that, **in order to improve a model’s reasoning ability while maintaining OOD performance, it is crucial to construct high-quality data from new data sources**. This highlights the effectiveness of our number sequence data and pipeline.
>
>
> **References**
>
> [5] Benchmarking Reasoning Capabilities of LLM Agents for Inductive Program Synthesis
>
> [6] Unveiling Over‑Memorization in Finetuning LLMs for Reasoning Tasks
>
> [7] The Pitfalls of Memorization
>
> [8] The Hyperfitting Phenomenon: Sharpening and Stabilizing LLMs for Open‑Ended Text Generation
>
> [9] Intern-S1: A Scientific Multimodal Foundation Model
>
> [10] SynLogic: Synthesizing Verifiable Reasoning Data at Scale for Learning Logical Reasoning and Beyond
>
> [11] Enigmata: Scaling Logical Reasoning in Large Language Models with Synthetic Verifiable Puzzles

---

### Official Review · Reviewer_L1Gb · 2025-11-10

**Soundness:** 3
**Presentation:** 4
**Contribution:** 4
**Rating:** 8
**Confidence:** 4

**Summary:**

The paper introduces **CodeSeq**, a novel synthetic *post-training* framework aimed at improving **inductive reasoning** capabilities in large language models (LLMs). Instead of relying on text-only reasoning data, CodeSeq leverages **number sequences** a domain that naturally encodes inductive structure and transforms them into *algorithmic reasoning problems* solvable by code.

The pipeline has three main stages:

1. **Sequence Algorithmization** – Number sequences are automatically converted into *code-driven tasks*, where the model must infer the generating rule and implement it as executable code. This enables **unit-test-based supervision**, providing objective correctness signals rather than fuzzy linguistic ones.

2. **Case-based Reflection Injection (SFT phase)** – Using two collaborating LLM agents (“working” and “guiding”), the system generates iterative *reflections*: failed test cases, diagnostic reasoning, and code revisions. These reflections are incorporated into supervised fine-tuning data, teaching the model to reason through *error detection and correction*.

3. **Solvability-estimated Reinforcement Learning (RL phase)** – The authors introduce a new reward, **CSSR (Case-Synergy Solvability Scaling Reward)**, which combines (a) the *difficulty* of a problem (estimated by its empirical solvability rate) and (b) the *quality* of the model’s generated test cases. This encourages exploration of harder inductive patterns while reinforcing effective self-checking behavior.

Experiments on LLaMA-3-8B and Qwen-2.5-7B show significant improvements on the proposed **General Term Generation (GTG)** benchmark (up to +42.7 points), as well as *transfer gains* on standard code reasoning datasets such as **HumanEval**, **MBPP**, and **LiveCodeBench**. Notably, performance on unrelated tasks (MMLU, BBH, Gaokao-Bench) remains stable, suggesting that the method enhances reasoning without harming broad language competence.

**Strengths:**

**Originality**

The paper is conceptually original in how it reimagines *inductive reasoning* training for LLMs through **code-driven number sequences** rather than conventional text or mathematical proofs. By operationalizing reasoning as a *code synthesis problem with verifiable test cases*, it bridges the gap between symbolic algorithm induction and neural post-training. The **Case-based Reflection Injection** stage adds further novelty teaching the model to reason through *error diagnosis and correction* in a structured manner. The introduction of the **CSSR (Case-Synergy Solvability Scaling Reward)** is also a creative step that makes reinforcement learning sensitive to task difficulty and model-generated validation quality. Together, these ideas represent a thoughtful blend of synthetic data generation, reflective supervision, and difficulty-aware RL.

**Quality**

The paper demonstrates methodological rigor and careful experimental design. The proposed framework is implemented systematically from automatic task generation to reflection-driven SFT and a clear RL finetuning phase. The experiments span two model families (LLaMA3 and Qwen2.5), multiple scales, and both *in-domain* (GTG) and *transfer* benchmarks (HumanEval, MBPP, LiveCodeBench). Ablations are included for reflection data, CSSR, and solvability thresholds, showing how each component contributes. The results are consistent and reproducible across settings, with transparent reporting of data volumes, model configurations, and cost. Importantly, the method demonstrates transferability rather than overfitting to the synthetic domain, implying strong generalization.

**Clarity**

The paper is clearly written and well-structured, with intuitive diagrams illustrating the CodeSeq pipeline. Each component ie sequence algorithmization, reflection injection, and RL is explained with sufficient algorithmic detail and examples. Mathematical notation for CSSR is compact yet interpretable, and pseudocode clarifies the data generation and training processes. The visual examples of reflection traces effectively communicate the iterative reasoning process the model learns. Even technically complex sections (like solvability estimation and reflection-guided data curation) are presented in an accessible way, supporting broad comprehension.

**Significance**

This work holds high significance for both *practical LLM improvement* and *conceptual progress* in reasoning research. Practically, it provides a scalable synthetic data pipeline that improves reasoning without relying on scarce human annotations. Conceptually, it advances the field’s understanding of how to align LLMs toward **self-verifying, case-based reasoning**, a crucial step beyond rote pattern prediction. The demonstrated transfer to open-ended code tasks shows the potential for CodeSeq to strengthen reasoning skills that generalize across domains. It also offers a reproducible testbed (the GTG benchmark) that can stimulate further research on inductive reasoning evaluation.

**Weaknesses:**

**1. Limited reasoning scope (narrow inductive domain)**

While number sequences offer a clean testbed for inductive reasoning, they represent a *narrow and low-entropy domain*. The model’s improved performance on these structured tasks may not necessarily translate to the kind of **open-domain or abstract reasoning** that motivates the paper. The inductive structure of integer sequences is fundamentally algorithmic and deterministic, which may inflate the appearance of “reasoning” without requiring conceptual generalization.

*Actionable insight:* Extend the pipeline to other symbolic or relational domains, e.g., **graph induction**, **analogy puzzles**, or **patterned logical expressions,** to test if the reflection mechanism generalizes beyond numeric patterns.


**2. Dependence on LLM-generated data introduces bias and noise**

The dataset is created using “working” and “guiding” LLM agents to generate both tasks and reflections. This **LLM-in-the-loop data generation** can embed the biases, style, and error distributions of the base models, producing self-reinforcing artifacts. There’s limited analysis of the diversity, correctness, or quality assurance of the generated reflections. The paper briefly mentions filtering invalid problems, but offers no systematic verification or human evaluation of data quality.

*Actionable insight:* Include a **human-audited sample analysis** (e.g., 100 random problems) quantifying correctness, novelty, and reflection coherence. Report statistics on how often guiding-agent feedback leads to genuinely correct fixes versus superficial edits.


**3. Evaluation contamination and reproducibility concerns**

The GTG benchmark used to demonstrate performance is internally generated by the authors from similar sources as the training data. The paper provides no detailed description of **data leakage prevention** or overlap detection between the training set and GTG test set. This raises potential issues of data contamination that could exaggerate model gains. Additionally, only aggregate metrics are presented; qualitative error analysis is minimal.

*Actionable insight:* Release scripts for overlap analysis or disclose specific measures (e.g., hash-based deduplication, source isolation). Include a **qualitative failure analysis** showing examples where models produce plausible but wrong rules, revealing what kind of reasoning errors persist.


**4. CSSR reward lacks theoretical justification and sensitivity analysis**

The proposed CSSR reward combines problem solvability and case success rate using a logarithmic scaling. While empirically useful, its **functional form is heuristic**, and its parameters (e.g., solvability cutoff 0.46, smoothing constant ε) are fixed without justification or robustness checks. The ablations show CSSR helps, but there’s no exploration of *why* or how sensitive results are to these values.

*Actionable insight:* Perform a **parameter sweep** on CSSR’s components or report confidence intervals to assess robustness. Provide a short theoretical or empirical rationale explaining why log-scaling solvability yields better exploration.


**5. Lack of deeper interpretive analysis of “reasoning” behavior**

Although the paper improves quantitative benchmarks, it doesn’t analyze *how* the model’s reasoning changes after CodeSeq training. There’s no introspection into whether the model actually learns iterative hypothesis testing or simply overfits to reflection patterns seen during SFT. The presented examples of reflection are anecdotal and do not evaluate correctness or consistency across reasoning steps.

*Actionable insight:* Include **behavioral or process-level evaluation**, such as reasoning trace consistency metrics or correlation between intermediate test-case success and final correctness. Even a small manual annotation of 50 reflections would reveal if the model internalizes a genuine error-checking habit or merely imitates the reflection style.


**6. Limited transparency and dataset accessibility**

Despite being a synthetic framework, the paper does not release datasets or reflection traces for reproducibility. The appendix mentions dataset statistics but omits exact sources, sample data, or implementation details for the reflection generator.

*Actionable insight:* Provide partial dataset releases or at least **10–20 illustrative full examples** of SFT and RL training data in the appendix. Releasing the GTG benchmark publicly would also enhance credibility and encourage further work on code-based induction.

**Questions:**

1. **On dataset overlap and contamination control**

   * Could you clarify how you ensured *no overlap* between the generated training data (from sequence algorithmization) and the **GTG test set**?
   * Were any hash-based deduplication, source separation, or semantic similarity filtering steps used to prevent data leakage?
   * A clear statement of contamination control would significantly strengthen the credibility of your evaluation, since both the training and test data derive from number-sequence sources.

2. **On the representativeness and quality of reflections**

   * How often do the *guiding-agent reflections* actually lead to *corrected solutions* versus superficial edits?
   * Did you perform any manual or automated quality check of reflection traces (e.g., proportion of cases where the fix was logically valid)?
   * Could you share statistics or examples showing the *distribution of reflection depth*—how many reasoning iterations typically occur before convergence?

3. **On CSSR reward design and robustness**

   * The CSSR reward function is defined heuristically as a log-scaled function of solvability combined with case-success rate. Could you provide an intuition or empirical justification for this formulation?
   * Why was the *solvability threshold* set to 0.46, and how sensitive are results to this value?
   * Have you tried linear or exponential alternatives to the log scaling, and what were the effects?
   * A short sensitivity or ablation study would help confirm that CSSR’s performance isn’t overly dependent on hand-tuned hyperparameters.

4. **On reasoning behavior and interpretability**

   * Do the models trained under CodeSeq show measurable changes in reasoning style? For instance, are their *reflection traces* or *self-debugging steps* more coherent or consistent compared to baseline models?
   * It would be helpful if you could include an *error typology* or a qualitative analysis of reflection chains, identifying whether models truly engage in hypothesis testing or merely imitate reflection syntax.
   * Could you provide examples of failure cases where the model’s reasoning remains brittle and discuss what these reveal about current limitations?

5. **On generalization beyond number sequences**

   * While number sequences are a good proxy for inductive structure, do you believe the CodeSeq pipeline generalizes to *non-numeric domains* such as symbolic reasoning, causal inference, or analogical tasks?
   * What modifications would be necessary for CodeSeq to operate on **graph patterns, textual analogies, or logic puzzles**?
   * If possible, please discuss one concrete obstacle or preliminary result showing whether the framework extends beyond algorithmic sequence induction.

6. **On data transparency and reproducibility**

   * Could you release at least a subset of the generated dataset—say, 100–200 SFT examples and reflection traces—to allow community validation and replication?
   * Will the **GTG benchmark** be made public, and if so, how do you plan to handle potential overlap with future training data in open checkpoints?
   * Publicly releasing even partial resources would substantially improve the paper’s impact and trustworthiness.

7. **On evaluation metrics and cross-domain effects**

   * Have you examined whether the CodeSeq-trained models demonstrate improved reasoning *process fidelity* (e.g., fewer self-contradictions, more accurate intermediate reasoning steps) rather than only higher end-task accuracy?
   * Additionally, could you expand on how you ensured that post-training on CodeSeq did not degrade performance on general-language reasoning benchmarks (e.g., MMLU, BBH)? Was any drop in specific reasoning categories observed?


**Overall suggestion**

Clarifications on dataset independence, reflection validity, and CSSR robustness would directly address the main limitations of the current version. Furthermore, presenting qualitative examples of reasoning progression and exploring possible extensions to other inductive domains would meaningfully strengthen the paper’s claim of advancing *true reasoning* in LLMs.

---

> ### Author Response · Authors · 2025-11-21
> **response to reviewer L1Gb #1**
>
> Thank you for your detailed and valuable comments!
>
> *For Weakness "reasoning scope" and Problem "generalization beyond number sequences"*:
> - Compared with other inductive data, number sequences exhibit **strong scalability** (a single sequence can theoretically contain infinitely many terms, each of which can be converted into a data point) and feature nested and progressive patterns, making them **more complex and closer to real-world scenarios**.
> - Precisely because of the high quality of number sequence data, they demonstrate **strong transfer performance on other inductive tasks**, such as ARC (Table 3) and List Function (Table A below).
>
> **Table A**
> |  | **ListFunction** |
> |-----------------|--------|
> | **LLaMA3-8B** | 0.17 |
> | **w/ CodeSeq**   | 0.23|
>
> - Our synthetic data can **be fully automatically generated and verified at a lower cost compared to other tasks**. For the types of inductive problems you mention, applying the same data pipeline may lead to several issues:
>   - Graph-structured data may not be easily convertible into textual forms familiar to LLMs [5], and graph rules cannot be executed and verified in the same straightforward way as code.
>   - Textual induction is susceptible to context, and multiple answers may be equally valid. Pure semantic matching is neither as rigorous nor as convenient as executable code for verification [6].
>   - Puzzle-style problems are feasible, but as puzzle complexity increases (e.g., in terms of the number of constraints or the scale of variables), the cost of expansion and verification rises significantly [7].
>
> *For Weakness "LLM-generated data introduces bias and noise"*:
> - Our pipeline **minimizes bias** as much as possible.
>   - Length bias. The length of a sequence is independent of the data: each term in the number sequence corresponds to one case in the algorithmic problem (Figure 1). We ensure that every problem contains exactly two example cases and 5 to 7 test cases, and this quantity is strictly controlled.
>   - Pattern bias. Each number sequence corresponds to a distinct inductive pattern. We ensure that each pattern appears an equal number in the training or test sets, preventing the base model or the test distribution from being skewed toward any particular pattern.
> - Although our pipeline involves LLMs, we **ensure data correctness through human-written Python rules to eliminate any noise** (Appendix L1219–L1231).
>   - The High-Density Sequence Data already includes background and mathematical descriptions, so the working agent only needs to wrap the descriptions into a story.
>   - For generated example cases, human-written rules check if they exist in the original number sequence and compute the offsets.
>   - A guiding agent generates answers from the example cases' input and problem descriptions, keeping only correct ones to ensure consistency between the problem descriptions and the example cases.
>   - To avoid bias or limited diversity, different agents are used for problem generation and solving. We sample each problem and its answer multiple times to ensure diversity.
> - We run all code in a **sandbox unit test independent of the LLM**, ensuring that every case executes correctly. This guarantees that the intermediate processes are noise-free and that the final ground-truth answers are completely accurate.
> - Through the rigorous steps described above, we can ensure consistency between the process and the results, which in turn allows us to:
>   - Guarantee that each problem is correct.
>   - Achieve rich diversity in problem samples (more than 2,500 number sequence patterns with various sampling strategies).
>   - Avoid superficial edits: the correctness of the final answers shows that the LLMs engage in a natural and proper reflection process.
>   - Retain the LLM’s natural reflection process, training LLMs to generate cases automatically.

---

> ### Author Response · Authors · 2025-11-21
> **response to reviewer L1Gb #2**
>
> *For Weakness "evaluation and reproducibility concerns"*:
> - There is no overlap between the training and test sets. Each number sequence represents a distinct inductive pattern and corresponds to a unique ID; **simply deduplicating by IDs is sufficient to prevent any overlap**.  Every data point in our SFT training set, RL training set, and GTG test set is unique with no repetition. As a result, the model encounters entirely new sequences during the testing phase.
> - Thank you for the reviewer’s comments. We conduct a statistical analysis and investigation on the cases where the Qwen2.5 model fails both before and after training.
>   - We find that the majority of failures (about 60%) are due to the general term formulas of the number sequences involving **compositional rules**. For example, the sequence 10, 3, 2, 2, 3, 3, 3, …, is generated by a formula that combines square root operations, recursion, and modulo operations.
>   - Another portion (about 30%) of the number sequences contains hidden variables, where the sequence alone is insufficient to infer the rule, requiring **reconstruction of the hidden state**. For instance, the sequence 1, 1, 2, 4, 3, 7, 5, 12, 7, … actually represents a robot moving in the directions right, down, left, and up in order, with each step’s displacement equal to the sum of the previous two steps. The general term of the sequence corresponds to the robot’s Manhattan distance (the hidden variable in this case) from the starting point.
>   - These kinds of patterns make it challenging for LLMs to learn.
>
> *For Weakness "CSSR reward" and Question "CSSR reward design and robustness"*:
> - Our reward function design comes from **findings in prior work**.
>   - The CSSR reward function consists of a solvability term with an additional case-specific reward term. [1] shows that if the additional reward exists in a weighted form, the theoretical optimal policy remains unchanged. On this basis, such multi-component rewards [2] are common and do not disrupt the fundamental optimization objective.
>   - In [3], the reward is constructed using historical success rates estimated from experience, and [4] demonstrates that applying a log function can stabilize training. Therefore, in the first term, we apply a log-based shaping operation to the solvability component.
> - The smoothing constant is used to **mathematically prevent the logarithm from being zero**.
> - In L1507–L1541 (Appendix A.5), we provide a detailed explanation of why we chose 0.46 as the cutoff. **Some experimental evidence support the value**. In summary:
>   - A cutoff of 0.46 means that problems with a pass rate below 15/32 are used for training, allowing us to select harder problems to maximize the model’s potential.
>   - Setting 0.46 as the threshold results in roughly 1.5K samples; exceeding this number would make the problems easier, and experiments show that easier training samples are not optimal (qwen2.5-7B on GTG task scores 0.65, less than the best 0.69).
>   - Experiments in Figure 3 demonstrate that, within the 0–0.46 range, more training data leads to better performance, so we use all data within this range.
> - We conduct **a detailed parameter sweep, component replacement, and sensitivity analysis for the CSSR function**.
>   - First, the results in Table 2 represent the test performance averaged over multiple training runs. We also train models of three different sizes (Table 2 and Table 9) to eliminate model-specific adaptation bias.
>   - Second, in Figures 3(b) and 3(c), we show the effects of different solvability ranges and different CSSR function forms (including linear mode) on the final reward curves, respectively.
>   - We have tried using an exponential function for the reward, i.e., in the form of $e^{-sov}$. After multiple runs and averaging the results, Qwen2.5-7B achieves around 0.66 on the GTG task, which is inferior to the performance obtained with the logarithmic formulation used in the paper.
>   - Finally, in Appendix A.7.4, we provide mathematical expressions for different reward function designs and further experimentally (Figure 3(c)) demonstrate that they cannot surpass the original design of the CSSR function.
>   - Taken together, these results demonstrate that our CSSR design is both optimal and robust.

---

> ### Author Response · Authors · 2025-11-21
> **response to reviewer L1Gb #3**
>
> *For Weakness "analysis of reasoning behavior"*:
> - The results in Figure 5 demonstrate that, after training, **the model can not only improve the correctness of constructing cases but also increase the final success rate in solving problems**. The correctness of the case construction in the model’s reflection traces and the accuracy of the generated hypotheses are consistent and positively correlated. This part of the experiment indicates that the model can, to some extent, learn to self-generate cases, thereby improving the hit rate of its generated hypotheses.
> - The SFT stage does not lead to overfitting because we strictly ensure that no number sequence is repeated across the train, and test sets. As a result, the model encounters entirely new number sequences during testing.
> - **The reflection process is strictly verified during reasoning**.
>   - Each reflection corresponds to a piece of code, which is executed and validated on cases through independent unit tests, regardless of the LLM. Therefore, the correctness of the code demonstrates the correctness of the reflection.
>   - If a reflection is correct, it produces the final correct code, which can be used as ground truth for training.
>   - If the code generated after reflection fails the unit tests, further reflection is conducted on the failing test cases until the final correct answer is obtained.
> - The reasoning trace is fully consistent with the final correctness, as the final result of the reasoning trace passes all test cases in independent unit tests. Because the correctness is verified externally, the reasoning outcome is deterministic, ensuring consistency between the process and the reasoning. Ultimately, **the error-checking behavior in reasoning is a natural behavior of the LLM, and these data are used to train the model to internalize this habit**.
>
> *For Weakness "dataset accessibility" and Problem "data transparency and reproducibility"*:
> - Thank you for the suggestion. However, we provide **an anonymous code repository** in Abstract, and we also include the training and test set in the **supplementary materials**.
> - The exact sources, sampling process with quantity details of the sample data, and implementation details mentioned by the reviewer can be found in Appendix A.4.1, Appendix A.5, and Appendix A.4(with Appendix A.6), respectively.
>
> *For Problem "dataset overlap and contamination control"*:
> - In "evaluation and reproducibility concerns", we use IDs to ensure that the training and test data contain no overlap and prevent data leakage.
> - As described in "LLM-generated data introduces bias and noise", we show how contamination is controlled during both data construction and evaluation.
>
> *For Problem "the representativeness and quality of reflections"*:
> - In Table 1, we report the frequency of using reflection to obtain the final correct result. In the training data, the model **performs 0 to 5 rounds of reflection, with an average of 2.56 rounds** of reflective reasoning needed to produce code that can pass all example tests. We present **the distribution of the number of reflection rounds** in Table 13.
> - In "analysis of reasoning behavior", we show how our reflection chains are strictly verified during reasoning, which prevents models from merely engaging in superficial imitation and trains models to internalize this habit.
>
> *For Problem "On reasoning behavior and interpretability"*:
> - As discussed in "analysis of reasoning behavior", Figure 5 shows that the accuracy of reflection traces of the base model is improved, and our reflection chains are strictly verified during reasoning, which prevents the model from merely engaging in superficial imitation.
> - We conduct an error analysis and categorization of failure cases in "evaluation and reproducibility concerns".
>
> *For Problem "evaluation metrics and cross-domain effects"*:
> - The experimental results in Figure 5 demonstrate that, after training, the model can not only improve the correctness of constructing cases but also increase the final success rate in solving problems. This indicates that our training data indeed improves the accuracy of intermediate reasoning steps.
> - The experimental results in Table 2 perfectly demonstrate the robustness of our data beyond inductive reasoning tasks. First, the model **shows improvements on close-domain coding tasks**. Second, the model **maintains its performance on OOD comprehensive reasoning tasks**, including MMLU and BBH.
>
> **References**
>
> [1] Policy invariance under reward transformations
>
> [2] Balancing Multiple Sources of Reward in Reinforcement Learning
>
> [3] Highly Efficient Self-Adaptive Reward Shaping for Reinforcement Learning
>
> [4] Using a Logarithmic Mapping to Enable Lower Discount Factors in Reinforcement Learning
>
> [5] GRAPHLLM: BOOSTING GRAPH REASONING ABILITY OF LARGE LANGUAGE MODEL
>
> [6] CodeScore: Evaluating Code Generation by Learning Code Execution
>
> [7] ZebraLogic: On the Scaling Limits of LLMs for Logical Reasoning

---

### Author Response · Authors · 2025-11-26
**response to all**

We sincerely thank all six reviewers for their valuable and insightful comments. We will incorporate the corresponding revisions in the updated manuscript to further enhance the completeness and rigor of our paper.

**Summary of revisions**
- Value of the number sequence data
  - We explain **why our data exhibits greater complexity** by analyzing the underlying rules and inductive structures. Furthermore, we highlight **the advantages of our data** over graph-based, linguistic, and visual datasets in terms of **scalability, verifiability, and construction efficiency**.
  - Using the same evaluation setting as the ARC task, we reveal that existing strong reasoning models perform poorly on number sequence tasks through experiments, further demonstrating **the research value of this type of data and that our method is not a simple distillation**.
- Correctness of the pipeline and algorithm
  - We strictly ensure **the correctness of the algorithmic problems** through the steps outlined in Appendix L1219–L1231. General-purpose code sandbox testing further guarantees **the correctness of code solutions** and prevents shortcuts that apply only to number sequences.
  - We explain how the **data pipeline avoids length and pattern biases** by transforming the number sequence terms into algorithmic cases and performing ID-based deduplication.
  - We **derive the idea behind the CSSR reward function** based on findings from prior work.
- Additional experiments
  - **To prove the transferability of our data to other inductive reasoning tasks**, we evaluate the Qwen2.5-7B model trained with our CodeSeq data on three OOD inductive reasoning tasks (ARC, ListFunctions, CodeArc). The results show the generalization ability of the data, which can enhance the model’s fundamental inductive reasoning capability.
  - **To prove the effectiveness of our data compared with other training datasets**, we train models separately using CodeSeq and other training data (Mirage and Humaneval), further demonstrating that our data can effectively raise the model’s performance ceiling.
  - **To explore the results of our training data on reasoning models**, we use Qwen2.5-coder-7B as the base model for the experiments in Table 2, and the results demonstrate that the data remains effective for reasoning models.
  - **To demonstrate the representativeness of our test set**, we expand the test set to the same scale as the training data (1,000). The results show that the performance difference between the two scales is minimal.
- More analyses
  - **To summarize the failure patterns for future research**, we analyse the failure cases and summarize two types of number sequence patterns that are particularly challenging for LLMs: compositional rules and reconstruction of the hidden state.

In order to further improve the paper, we sincerely hope to receive the reviewers’ feedback as soon as possible.

---

### Author Response · Authors · 2025-11-27
**Gentle Reminder: Any Further Thoughts?**

Dear Reviewers,

I hope this message finds you well. As the discussion period is nearing its end with about one week left, I want to ensure we have addressed all your concerns satisfactorily. If there are any additional points or feedback you'd like us to consider, please let us know. Your insights are invaluable to us, and we are eager to address any remaining issues to improve our work.

Thank you for your time and effort in reviewing our paper.

---

### Meta-Review · Area_Chair_mE4E · 2025-12-20

**Summary:**

The paper proposes CodeSeq, a synthetic data pipeline utilizing number sequences and code generation to enhance the inductive reasoning capabilities of LLMs. While reviewers recognized the soundness of the construction pipeline and the novelty of the verification mechanism, it is questionable about whether number sequence dataset can indeed improve general inductive reasoning. Crucially, despite additional experiments, the method failed to demonstrate significant transfer gains on established, challenging benchmarks like ARC. The skepticism regarding the method's practical significance and broader generalization remains unresolved. Therefore, I do not recommend acceptance for this paper.

**Reviewer Concerns:**

Addressed:
1. Since number sequences are scraped, models might have memorized them. The authors clarified they use ID-based deduplication to ensure zero overlap between train/test.
2. The test set of 200 samples was too small. The authors expanded the test set to 1,000 samples.
3. Unclear costs and lack of failure analysis. The authors provided a specific cost breakdown and a detailed failure analysis

Outstanding:
1. Reviewers were not convinced that solving number sequences is the right proxy for general inductive reasoning.
2. The ability learned using the proposed dataset cannot be transfered to other problems like ARC.

**Reviewer Scores:**

Reviewer L1Gb may not change the score because it's already 8.
Reviewer Evb6 said he/she will increase to 4, but still remain sceptical.
Reviewer iUqk, aQXx and FLvu may not change the score because his central question about whether sequence data represents inductive reasoning is not resolved.
Reviewer kxad is unclear.

---

### Decision · Program_Chairs · 2026-01-26

Reject